# Whoever Started the Interference Should End It: Guiding Data-Free Model Merging via Task Vectors

**Runxi Cheng** [1] [*]   **Feng Xiong** [2] [*]   **Yongxian Wei** [1]   **Wanyun Zhu** [3]   **Chun Yuan** [1] [†]

## Abstract

Model merging seeks to integrate task-specific expert models into a unified architecture while preserving multi-task generalization capabilities, yet parameter interference between constituent models frequently induces performance degradation. Although prior work has explored many merging strategies, resolving interference without additional data for retraining or test-time computation remains challenging. In this paper, we theoretically demonstrate that the task vectors of the linear layer constitute an approximate linear subspace for its corresponding input. Therefore, we can minimize interference under the guidance of task vectors. Based on this insight, we propose **WUDI-Merging** (**W**hoever started the interference sho**U**ld en**D I**t), a simple yet effective model merging method that eliminates interference without any additional data or rescaling coefficients. Comprehensive empirical evaluations across vision and language benchmarks demonstrate our method's superiority, achieving state-of-the-art performance in data-free model merging scenarios (average 10.9% improvement versus baseline methods) while even outperforming mainstream test-time adaptation approaches by 3.3%, and only very few computing resources are required. The source code and implementation details are available at https://github.com/nathanielyvo/WUDI-Merging.

[*]Equal contribution   [1]Tsinghua Shenzhen International Graduate School, Tsinghua University [2]Guangdong Provincial Key Laboratory of Novel Security Intelligence Technologies [3]CUHK-Shenzhen. Correspondence to: Runxi Cheng <crx23@mails.tsinghua.edu.cn>, Chun Yuan <yuanc@sz.tsinghua.edu.cn>.

*Proceedings of the $42^{nd}$ International Conference on Machine Learning*, Vancouver, Canada. PMLR 267, 2025. Copyright 2025 by the author(s).

## 1. Introduction

With the widespread adoption of the pre-training and fine-tuning paradigm, a large number of pre-trained and fine-tuned checkpoints have been released in open-source communities. However, directly applying multiple individual models for multi-task problems incurs significant storage costs. While multi-task learning has been employed to address this issue, it typically requires costly training and poses potential privacy risks. Recently, *model merging* has emerged as a solution, enabling the integration of multiple expert models into a single unified multi-task model without the need for expensive retraining on multi-task datasets.

However, due to interference among the expert models, the merged model exhibits a performance gap when compared to its corresponding expert models on specific tasks. Numerous model merging approaches have been proposed to address this issue. *Test-time adaptation* model merging methods, such as AdaMerging (Yang et al., 2024b) and Surgery (Yang et al., 2024a), leverage unlabeled test data to resolve interference through reweighting or model editing. *MoE-like* (Mixture-of-Experts) model merging methods, exemplified by EMR-Merging (Huang et al., 2024), retain additional task-specific parameters, such as task-specific masks, to prevent interference when integrating diverse expert models into a unified model. While these approaches effectively mitigate the interference, they require extra storage or access to test data. Conversely, *data-free* model merging methods, like Task Arithmetic (Ilharco et al., 2023), enable model merging without additional data or storage requirements, which is particularly advantageous when data privacy or availability is a concern. However, due to the absence of data, current data-free model merging techniques still exhibit a performance gap compared to the aforementioned methods and multi-task learning.

In this paper, we revisit the update process of fine-tuning phase. Consider a linear layer whose parameters are updated through gradient descent: each neuron's weight adjustment derives from the product of the learning rate, the gradient of its output, and the corresponding input vector. Crucially, since the learning rate and total optimization steps always remain constrained during standard fine-tuning procedures, the corresponding inputs for individual samples remain con-

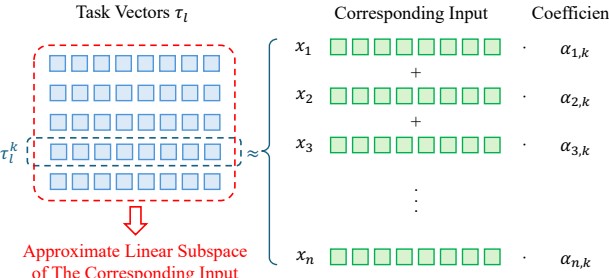

Figure 1: The task vector constitue an approximate linear subspace of its corresponding input.

sistent across successive update iterations. The temporal consistency indicated that the cumulative update to each neuron can be approximated as a weighted summation of fixed input vectors, with coefficients determined by the product of learning rates and corresponding gradient magnitudes. This implies that the task vectors of a linear layer constitute an approximate linear subspace of the input, allowing us to implicitly utilize training data information solely through the task vectors. In this context, the task vectors are computed by calculating the difference between the weights of the expert models and those of the pre-trained model.

Based on this insight, we propose **WUDI-Merging** (**W**hoever started the interference sho**U**d en**D** **I**t), a simple yet effective data-free model merging method. We evaluate our method on both vision and natural language processing tasks, where our method significantly surpasses recent data-free model merging techniques and outperforms mainstream test-time adaptation model merging methods. In Summary, our contributions are summarized as follows:

- Through detailed theoretical analysis, we have demonstrated that the task vectors within the linear layer approximately form a linear subspace of the input space.

- We propose WUDI-Merging, a simple yet effective data-free model merging method that minimizes interference to the task vector without requiring any additional data, extra storage, or rescaling coefficients.

- Extensive evaluations have demonstrated that WUDI-Merging achieves state-of-the-art performance in data-free model merging, surpassing mainstream test-time adaptation model merging methods.

## 2. Related Work

Current research primarily falls into three categories: Data-free, Test-time adaption, and MoE-like methods.

**Data-Free Model Merging:** Data-free model merging aims to combine different expert models without any additional data for retraining. Wortsman et al. (2022) proposed *Simple Averaging*, which constructs the merged model by averaging the parameters across all models. Matena & Raffel (2022)

introduced *Fisher Merging*, performing weighted model merging by utilizing the Fisher information matrix to assess the importance coefficient of expert models. Jin et al. (2023) addressed model merging by minimizing prediction differences between the merged model and the expert models. Recently, Ilharco et al. (2023) demonstrated that we can effectively edit models by simply applying arithmetic operations to task vectors. However, the interference among expert models remains a significant challenge. Yadav et al. (2023) proposed *Ties-Merging* to resolve this challenge by three novel steps. Yu et al. (2024) introduced *DARE*, which resolves the merging interference by dropping parameters and unscaling operations. Wang et al. (2024) introduced *Consensus Merging*, which enhanced the overall performance of existing model merging methods by removing the selfish and catastrophic weights. Furthermore, Xiong et al. (2024); Wei et al. (2025) proposed to align the loss between the merged model and expert models. Although these methods alleviate interference between experts to some extent, a gap still exists between data-free model merging and test-time adaptation model merging methods.

**Test-time adaptation Model Merging:** Test-time adaptation model merging leverages a portion of unlabeled test data to resolve interference among expert models. Yang et al. (2024b) proposed *AdaMerging*, which utilized entropy minimization on unlabeled test samples as a heuristic objective function to learn the merging coefficients. In addition, Yang et al. (2024a) introduced *Representation Surgery*, which reduces representation bias by minimizing the distance between the representations of the merged model and those of individual models. Despite these methods are promising, a critical limitation of these methods lies in their reliance on access to test data, which may not be feasible in practice.

**MoE-like Model Merging:** MoE-like model merging has emerged as a promising paradigm for leveraging multiple specialized models while storing task-specific knowledge in multi-task learning scenarios. Huang et al. (2024) proposed *EMR-Merging*, which elect a unified model and uses task-specific masks for multi-task learning problems. Lu et al. (2024) introduced *Twin-Merging*, incorporating a router to dynamically merge shared and exclusive knowledge based on the test inputs. While these methods effectively mitigate interference issues in model merging, they present certain practical limitations. The requirement for storing multiple task-specific components increases memory overhead.

Although test-time adaptation and MoE-like methods have achieved remarkable results in resolving interference among expert models, their practical applicability is limited. This limitation arises from challenges such as data privacy concerns, the need for extra storage, and the lack of parallelism in unified models. Therefore, in this paper, we primarily focus on the data-free model merging.

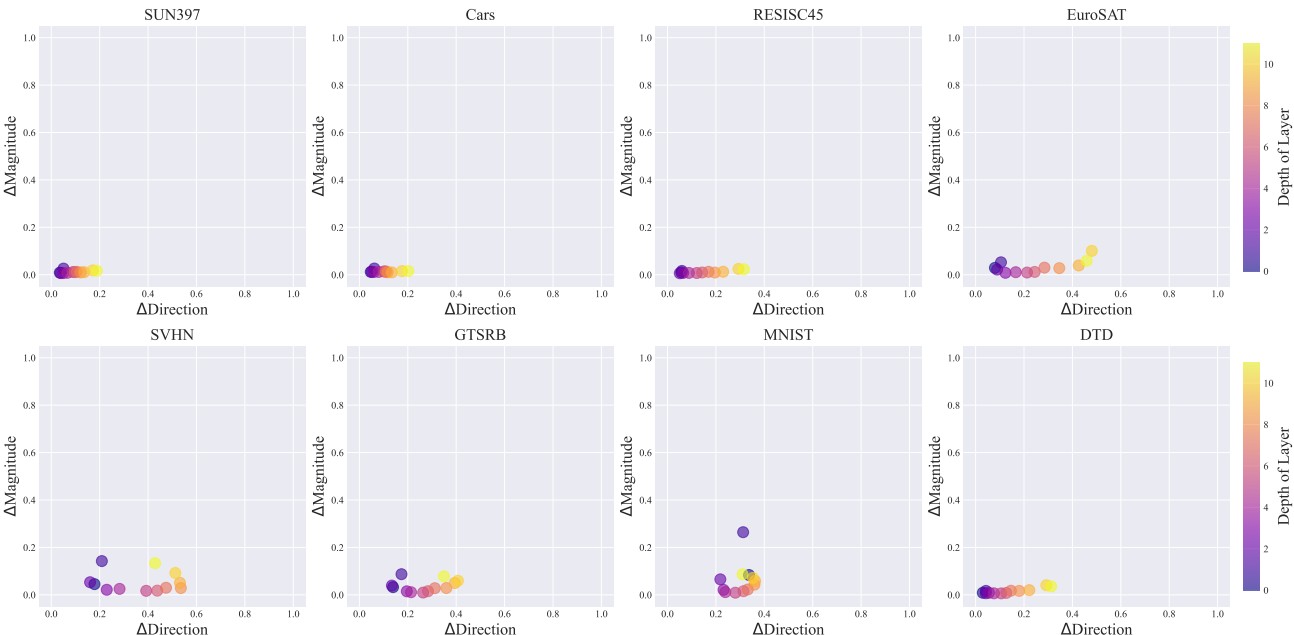

Figure 2: The input consistency between the pretrained model and the fine-tuned model for ViT-B/32

## 3. Methodology

### 3.1. Preliminary of Model Merging

**Notation.** Formally, let $\boldsymbol{\theta}$ denote the parameters of the pretrained model, and let $\boldsymbol{\theta}_i$ represent the parameters of the expert model for task $i$, fine-tuned from $\boldsymbol{\theta}$. Following Task Arithmetic (Ilharco et al., 2023), we define the *task vector* $\boldsymbol{\tau}_i$ for task $i$ as the difference between the expert model and the pretrained model:

$$\boldsymbol{\tau}_i = \boldsymbol{\theta}_i - \boldsymbol{\theta} \tag{1}$$

We then define the merged task vector as $\boldsymbol{\tau}_m$, and the final parameters of the merged model as $\boldsymbol{\theta}_m = \boldsymbol{\theta} + \boldsymbol{\tau}_m$

**The impact of the Linear Layer's Task Vector.** As demonstrated by Jin et al. (2023); Xiong et al. (2024), the task vector in the linear layer encapsulates most of the capabilities of the expert models. As shown in Figure A.3, an expert model utilizing only the task vector of the linear layer achieves performance comparable to that of the full expert model. Therefore, *we primarily focus on the linear layers of the model.*

**Interference Vector.** Empirical observation (Ilharco et al., 2023) reveals that task vectors for a certain pretrained model and task exhibit consistent directional characteristics within the parameter manifold. This directional consistency suggests that task vectors converge to local optima. Therefore, for task $i$, we defined the rest of the weight in $\boldsymbol{\tau}_m$ as interference vector $\boldsymbol{\delta}_i$, which is formulated as:

$$\boldsymbol{\delta}_i = \boldsymbol{\tau}_m - \boldsymbol{\tau}_i \tag{2}$$

**Interference of the Linear Layer.** Let $\boldsymbol{\theta}_{i,l}, \boldsymbol{\theta}_{m,l}$ denote the parameters for the linear layer $l$ of expert model for task $i$ and merged model, respectively. Similarly, let $\boldsymbol{\tau}_{i,l}$ and $\boldsymbol{\tau}_{m,l}$ represent the corresponding task vectors associated with $\boldsymbol{\theta}_{i,l}$ and $\boldsymbol{\theta}_{m,l}$. Additionally, we define corresponding input as $\boldsymbol{x}_{i,l}$. Then, we can formally define the interference of the $l$-th linear linear for task $i$ as follows (Jin et al., 2023; Fang et al., 2024; Zeng et al., 2019):

$$\mathcal{J}_i(\boldsymbol{\tau}_{m,l}) = \mathbb{E}_{\boldsymbol{x}_{i,l} \sim p(\boldsymbol{x}_{i,l})} \|\boldsymbol{\theta}_{m,l}\boldsymbol{x}_{i,l} - \boldsymbol{\theta}_{i,l}\boldsymbol{x}_{i,l}\|_2^2 \tag{3}$$

$$= \mathbb{E}_{\boldsymbol{x}_{i,l} \sim p(\boldsymbol{x}_{i,l})} \|\boldsymbol{\tau}_{m,l}\boldsymbol{x}_{i,l} - \boldsymbol{\tau}_{i,l}\boldsymbol{x}_{i,l}\|_2^2 \tag{4}$$

$$= \mathbb{E}_{\boldsymbol{x}_{i,l} \sim p(\boldsymbol{x}_{i,l})} \|\boldsymbol{\delta}_{i,l}\boldsymbol{x}_{i,l}\|_2^2 \tag{5}$$

### 3.2. Towards Understanding the Task Vector

In practical fine-tuning scenarios, the process typically employs a small learning rate and is executed over a limited number of iterations and we assume the model is Lipschitz continuous (Kim et al., 2021; Latorre et al., 2020; Fazlyab et al., 2019). Based on these, we propose Lemma 1 to analyze the consistency of the input for the linear layer $l$.

**Lemma 1** (Input consistency)**.** *Let $\boldsymbol{x}_l^p, \boldsymbol{x}_l^q$ denote the input of the linear layer $l$ after $p$ and $q$ $(p > q)$ iterations during fine-tuning, respectively. Denote $\eta_i$ as the learning rate of iteration $i$. Assume that the model before layer $l$ is $\mathcal{C}_l$-Lipschitz continuous, and the gradient of loss is bounded in $\ell_2$-norm by $\mathcal{G}_l$, then:*

$$\|\boldsymbol{x}_l^p - \boldsymbol{x}_l^q\|_2 \leq \mathcal{C}_l\mathcal{G}_l(\textstyle\sum_{i=q+1}^p \eta_i) \tag{6}$$

*The proof is in Appendix A.1.1.*

According to Lemma 1, the input of a individual sample to linear layer $l$ remain highly consistent across different iterations when both the learning rate and the number of iterations are small. We further validate this conclusion through an experimental study. Specifically, we compute the input consistency between the pre-trained model and the fine-tuned model, where the input of the pre-trained model can be seen as $x_l^0$ and the input of the expert model can be seen as the $x_l^T$.

$$\Delta\text{Direction} = 1 - \cos(\boldsymbol{x}_{\text{expert}}, \boldsymbol{x}_{\text{pretrain}}) \quad (7)$$

$$\Delta\text{Magnitude} = \frac{|\|\boldsymbol{x}_{\text{expert}}\|_2 - \|\boldsymbol{x}_{\text{pretrain}}\|_2|}{\|\boldsymbol{x}_{\text{pretrain}}\|_2} \quad (8)$$

Where the $\boldsymbol{x}_{\text{expert}}$ and $\boldsymbol{x}_{\text{pretrain}}$ represent the input of the expert model and the pretrained model, respectively. The result is shown in Fig. 2. Most layers exhibit high consistency, with direction changes less than 0.4, indicating that the inputs remains highly consistency between the pretrained and fine-tuned models. With this observation, we reconsider the update process of the task vector, which can be formulated as follows:

$$\boldsymbol{\tau}_l^k = \sum_{t=1}^{\mathrm{T}} -\eta_t \cdot \frac{\partial \mathcal{L}(\boldsymbol{\theta}^{t-1})}{\partial \boldsymbol{\theta}_{l,k}^{t-1}} \quad (9)$$

$$= \sum_{t=1}^{\mathrm{T}} -\eta_t \sum_{n=1}^{N} \frac{\partial \mathcal{L}(\boldsymbol{\theta}^{t-1})}{\partial(\boldsymbol{\theta}_{l,k}^{t-1}\boldsymbol{x}_{n,l}^{t-1})} \cdot \frac{\partial(\boldsymbol{\theta}_{l,k}^{t-1}\boldsymbol{x}_{n,l}^{t-1})}{\partial\boldsymbol{\theta}_{l,k}^{t-1}} \quad (10)$$

$$= \underbrace{\sum_{t=1}^{\mathrm{T}} -\eta_t \sum_{n=1}^{N} \frac{\partial \mathcal{L}(\boldsymbol{\theta}^{t-1})}{\partial(\boldsymbol{\theta}_{l,k}^{t-1}\boldsymbol{x}_{n,l}^{t-1})}}_{\text{coefficient}} \cdot (\boldsymbol{x}_{n,l}^{t-1})^{\top} \quad (11)$$

where $\boldsymbol{\tau}_l^k$ denote the task vector of neuron $k$ in linear layer $l$, and $\boldsymbol{\theta}_{l,k}^{t-1}$ denote the parameters of neuron $k$ in linear layer $l$ at time $t$. According to Eq. 11, each neuron in the linear layer can be interpreted as a weighted sum of the inputs across different iterations. Given that the inputs remain highly consistent across iterations, as previously established, we propose Proposition 1:

**Proposition 1** (Approximate Linear Combination). *Let $\boldsymbol{\tau}_l^k$ denote the task vector of neuron $k$ in linear layer $l$. Consider $N$ input samples $\{\boldsymbol{x}_n\}_{n=1}^N$, and let $\boldsymbol{x}_{n,l}^t$ denote the corresponding input to layer $l$ after $t$ iteration for sample $\boldsymbol{x}_n$ during fine-tuning. Assume the gradient of the loss with respect to the product $\boldsymbol{\tau}_l^k\boldsymbol{x}_{n,l}^t$ is bounded by $\Gamma_l$. Then, the following inequality holds:*

$$\left\| \boldsymbol{\tau}_l^k - \sum_{n=1}^{N} \beta_{n,l}^k(\boldsymbol{x}_{n,l}^{\mathrm{T}})^{\top} \right\|_2 \le \Phi_l \cdot (\sum_{t=1}^{\mathrm{T}} \sum_{i=t}^{\mathrm{T}} \eta_t\eta_i). \quad (12)$$

*where $\Phi_l = N \cdot \mathcal{C}_l \cdot \mathcal{G}_l \cdot \Gamma_l$ and $\beta_{n,l}^k = \sum_{t=1}^{\mathrm{T}} -\eta_t \frac{\partial \mathcal{L}(\boldsymbol{\theta}^t)}{\partial(\boldsymbol{\theta}_{l,k}^{t-1}\boldsymbol{x}_{n,l}^t)}$. The proof is in Appendix A.1.2.*

Consequently, each neuron in the linear layer can be approximated as a weighted sum of its corresponding inputs. That is to say, for linear layer $l$, the task vector $\boldsymbol{\tau}_{i,l}$ constitutes

an approximate linear subspace of its associated inputs. By leveraging this property, we can exploit the training data information solely through the task vectors.

### 3.3. Methodology

In this section, we introduce WUDI-Merging (**W**hoever started the interference sho**U**d en**D** **I**t), a simple yet effective Data-Free Model Merging method that operates without the need for any additional data or rescaling coefficients. Building upon the insights from Section 3.2, we observe that the task vector constitutes an approximate linear subspace for its corresponding inputs. This property motivated us to reconstruct the inputs by the task vectors:

$$\boldsymbol{x}_{i,l} = \sum_{k=1}^{K} \alpha_{i,l}^k(\boldsymbol{x}_{i,l})(\boldsymbol{\tau}_{i,l}^k)^{\top} + \boldsymbol{\varepsilon}(\boldsymbol{x}_{i,l}), \quad \boldsymbol{x}_{i,l} \in \mathcal{D}_{i,l} \quad (13)$$

Where $\alpha_{i,l}^k(\boldsymbol{x}_{i,l})$ are the reconstruction coefficients of $\boldsymbol{x}_{i,l}$, and $\boldsymbol{\varepsilon}(\boldsymbol{x}_{i,l})$ represents the reconstruction error. Then, we can reformulate the interference of the linear layer $l$ as follows:

$$\mathcal{J}_i(\boldsymbol{\tau}_{m,l}) = \mathbb{E}_{\boldsymbol{x}_{i,l}} \|\boldsymbol{\delta}_{i,l}(\sum_{k=1}^{K} \alpha_{i,l}^k(\boldsymbol{x}_{i,l})(\boldsymbol{\tau}_{i,l}^k)^{\top} + \boldsymbol{\varepsilon}(\boldsymbol{x}_{i,l}))\|_2^2 \quad (14)$$

Based on these discussions, we propose Theorem 1:

**Theorem 1** (Upper bound of interference). *Denote $\boldsymbol{x}_{i,l}$ as the linear layer $l$'s input of task $i$, $\boldsymbol{\delta}_{i,l}$ and $\boldsymbol{\tau}_{i,l}$ as the linear layer $l$'s interference vector and task vector of task $i$, respectively, then:*

$$\mathbb{E}_{\boldsymbol{x}_{i,l}\sim p(\boldsymbol{x}_{i,l})} \|\boldsymbol{\delta}_{i,l}\boldsymbol{x}_{i,l}\|_2^2 \le \omega_{i,l}^1 \cdot \left\| \boldsymbol{\delta}_{i,l}(\boldsymbol{\tau}_{i,l})^{\top} \right\|_F^2 + \omega_{i,l}^2 \cdot \|\boldsymbol{\delta}_{i,l}\|_F^2 \quad (15)$$

*Where $\omega_{i,l}^1$ and $\omega_{i,l}^2$ is the reconstruction constant. The proof is in Appendix A.1.3.*

According to Theorem 1, we can minimize the upper bound of interference for task $i$ by:

$$\min_{\boldsymbol{\delta}_{i,l}} \omega_{i,l}^1 \cdot \left\| \boldsymbol{\delta}_{i,l}(\boldsymbol{\tau}_{i,l})^{\top} \right\|_F^2 + \omega_{i,l}^2 \cdot \|\boldsymbol{\delta}_{i,l}\|_F^2 \quad (16)$$

$$\Leftrightarrow \min_{\boldsymbol{\delta}_{i,l}} \left\| \boldsymbol{\delta}_{i,l}(\boldsymbol{\tau}_{i,l})^{\top} \right\|_F^2 + \frac{\omega_{i,l}^2}{\omega_{i,l}^1} \cdot \|\boldsymbol{\delta}_{i,l}\|_F^2 \quad (17)$$

However, the reconstruction constant is related to the actual input, which is inaccessible. As a substitute, we can use an empirical coefficient $\omega$. Consequently, we consider the following optimization objective:

$$\min_{\boldsymbol{\delta}_{i,l}} \left\| \boldsymbol{\delta}_{i,l}(\boldsymbol{\tau}_{i,l})^{\top} \right\|_F^2 + \omega \cdot \|\boldsymbol{\delta}_{i,l}\|_F^2 \quad (18)$$

Given that task vectors of different scales can tolerate varying degrees of deviation, we adjust our optimization accordingly. Specifically, we weight the loss of each task based

on the scale of its task vector. By weighting each loss with the square of the F-norm, we balance the interference of different tasks. Therefore, the final objective function can be formulated as:

$$\min_{\{\boldsymbol{\delta}_{i,l}\}} \sum_i \frac{1}{\|\boldsymbol{\tau}_{i,l}\|_F^2}(\|\boldsymbol{\delta}_{i,l}(\boldsymbol{\tau}_{i,l})^\top\|_F^2 + \omega \cdot \|\boldsymbol{\delta}_{i,l}\|_F^2)$$

$$\Leftrightarrow \min_{\boldsymbol{\tau}_{m,l}} \sum_i \frac{1}{\|\boldsymbol{\tau}_{i,l}\|_F^2}(\|(\boldsymbol{\tau}_{m,l} - \boldsymbol{\tau}_{i,l})(\boldsymbol{\tau}_{i,l})^\top\|_F^2$$

$$+ \omega \cdot \|\boldsymbol{\tau}_{m,l} - \boldsymbol{\tau}_{i,l}\|_F^2) \quad (19)$$

Then, we can get the close-form solution for Eq. 19,

$$\boldsymbol{\tau}_{m,l} = \text{Matmul}(\sum_i \frac{1}{\|\boldsymbol{\tau}_{i,l}\|_F^2}\boldsymbol{\tau}_{i,l}(\boldsymbol{\tau}_{i,l}^\top\boldsymbol{\tau}_{i,l} + \omega I),$$

$$(\sum_i \frac{1}{\|\boldsymbol{\tau}_{i,l}\|_F^2}(\boldsymbol{\tau}_{i,l}^\top\boldsymbol{\tau}_{i,l} + \omega I))^{-1}) \quad (20)$$

The detailed derivation is in Appendix A.1.4. However, selecting a general regularization coefficient $\omega$ suitable for different tasks is challenging. To address this issue, we propose an alternative approach. Revisiting Eq.18, which can be seen as a form of ridge regression, and motivated by prior work (Smith et al., 2021; Wang et al., 2022; Zou et al., 2021) which have demonstrated that gradient descent methods such as SGD and Adam induce implicit regularization and shown better generalization in such problems, we propose to solve the problem by using the gradient descent. Therefore, we can eliminate the regularization term, which allows us to avoid the need to search for an appropriate coefficient. Accordingly, the loss function for gradient descent is formulated as follows:

$$\mathcal{L}_l = \sum_i \frac{1}{\|\boldsymbol{\tau}_{i,l}\|_F^2} \|(\boldsymbol{\tau}_{m,l} - \boldsymbol{\tau}_{i,l})(\boldsymbol{\tau}_{i,l})^\top\|_F^2 \quad (21)$$

Since the optimization of each linear layer is independent, we can sequentially solve the problem layer by layer, thereby avoiding significant computational overhead. The algorithmic flow is detailed in Algorithm 1.

# 4. Experiment

## 4.1. Experimental Settings

**Datasets.** For *vision* tasks, following (Yang et al., 2024b; Wang et al., 2024), we investigate multi-task model merging across eight image classification datasets. For *discriminative language tasks*, we employ the GLUE (Wang et al., 2018) to assess our method. For *generative language tasks*, following Yu et al. (2024), we evaluate our method on instruction-following, mathematical reasoning and code-generation.

**Models.** For *vision* tasks, we employ the ViTs (Dosovitskiy et al., 2020) derived from CLIP (Radford et al., 2021), including ViT-B/32, ViT-B/16 and ViT-L/14. For *discriminative language* tasks, we employ the RoBERTa-Base (Liu et al., 2019) and RoBERTa-Large for evaluation. For *generative language* tasks, we employ the Llama2 (Touvron et al., 2023) for evaluation.

---

**Algorithm 1** WUDI-Merging

1: **Input:** pretrained model parameters $\boldsymbol{\theta}$; task vectors $\mathcal{T} = \{\boldsymbol{\tau}_{i,l}\}_{i=1}^{\mathcal{P}}$; solution steps $\mathcal{N}$; learning rate $\zeta$.
2: **Output:** merged multi-task model parameters $\boldsymbol{\theta_m}$.
3: ▷ Initialize merged task vector: $\boldsymbol{\tau}_{m,l}^0 = \sum_i \boldsymbol{\tau}_{i,l}$.
4: **for** linear layer $l \in \{1, \cdots, \Psi\}$ **do**
5:    **for** $n \in \{1, \cdots, \mathcal{N}\}$ **do**
6:       ▷ Calculate loss by merged task vector and task vectors:
7:       $\mathcal{L}_l = \sum_i \frac{1}{\|\boldsymbol{\tau}_{i,l}\|_F^2} \|(\boldsymbol{\tau}_{m,l} - \boldsymbol{\tau}_{i,l})(\boldsymbol{\tau}_{i,l})^\top\|_F^2$
8:       ▷ Update the merged task vector $\boldsymbol{\tau}_{m,l}^n$:
9:       $\boldsymbol{\tau}_{m,l}^n = \boldsymbol{\tau}_{m,l}^{n-1} - \zeta \cdot \nabla_{\boldsymbol{\tau}_{m,l}^{n-1}}\mathcal{L}_l\left(\boldsymbol{\tau}_{m,l}^{n-1}; \mathcal{T}\right)$
10:    **end for**
11: **end for**
12: ▷ Assemble the merged task vectors from all linear layers:
13: $\boldsymbol{\tau}_m = \{\boldsymbol{\tau}_{m,l}^{\mathcal{N}}\}_{l=1}^{\Psi}$
14: ▷ Calculate the Merged multi-task model parameters:
15: $\boldsymbol{\theta_m} = \boldsymbol{\theta} + \boldsymbol{\tau}_m$

---

**Baselines:** We primarily compared WUDI-Merging with recent data-free model merging methods, including Weight Averaging (Wortsman et al., 2022), Fisher Merging (Matena & Raffel, 2022), RegMean (Jin et al., 2023), Task Arithmetic (Ilharco et al., 2023), Ties-Merging (Yadav et al., 2023), Consensus Merging (Wang et al., 2024), and PCB Merging (Du et al., 2024). Additionally, we compared WUDI-Merging with mainstream test-time adaptation methods, namely Adamerging (Yang et al., 2024b) and Surgery (Yang et al., 2024a).

**Experiment Details.** Our method only uses two hyperparameters. In practice, we used Adam optimizer and set the learning rate to 1e-5. The number of iterations is set to 300. Following (Jin et al., 2023; Xiong et al., 2024), we only applied our method to the linear layer in the model. We provide more details in A.8

## 4.2. Main Results

**Results of Visual Tasks.** The main experimental results of vision tasks are presented in Tables 1, 2, and 10. These findings illustrate that a performance gap generally persists between Test-Time Adaptation (TTA) methods and current data-free methods. Compared to the aforementioned approaches, our proposed method notably outperforms all existing methods without utilizing additional data or requiring extra storage. Specifically, on the ViT-B/32 model, WUDI-Merging achieves 8.9% performance improvement over the sota data-free method PCB-Merging and surpasses the leading TTA method AdaMerging++ by 4.1%. For the ViT-L/14 model, WUDI-Merging exceeds PCB-Merging by 5.1% and outperforms AdaMerging++ by 1.6%. Notably,

Table 1: Multi-task performance when merging ViT-B/32 models on 8-task vision benchmark.

| Method | SUN397 | Cars | RESISC45 | EuroSAT | SVHN | GTSRB | MNIST | DTD | Avg Acc |
|---|---|---|---|---|---|---|---|---|---|
| *Non-merging Methods* | | | | | | | | | |
| Pretrained | 62.3 | 59.7 | 60.7 | 45.5 | 31.4 | 32.6 | 48.5 | 43.8 | 48.0 |
| Individual | 79.2 | 77.7 | 96.1 | 99.7 | 97.5 | 98.7 | 99.7 | 79.4 | 90.8 |
| Traditional MTL | 73.9 | 74.4 | 93.9 | 98.2 | 95.8 | 98.9 | 99.5 | 77.9 | 88.9 |
| *Test-time Adaptation Methods* | | | | | | | | | |
| AdaMerging | 64.5 | 68.1 | 79.2 | 93.8 | 87.0 | 91.9 | 97.5 | 59.1 | 80.1 |
| AdaMerging++ | 66.6 | 68.3 | 82.2 | 94.2 | 89.6 | 89.0 | 98.3 | 60.6 | 81.1 |
| Representation Surgery | 63.8 | 59.9 | 83.3 | 97.9 | 87.0 | 87.0 | 98.6 | 69.4 | 80.9 |
| *Data-free Methods* | | | | | | | | | |
| Weight Averaging | 65.3 | 63.4 | 71.4 | 71.7 | 64.2 | 52.8 | 87.5 | 50.1 | 65.8 |
| Fisher Merging | 68.6 | 69.2 | 70.7 | 66.4 | 72.9 | 51.1 | 87.9 | 59.9 | 68.3 |
| RegMean | 65.3 | 63.5 | 75.6 | 78.6 | 78.1 | 67.4 | 93.7 | 52.0 | 71.8 |
| Task Arithmetic | 55.2 | 54.9 | 66.7 | 78.9 | 80.2 | 69.7 | 97.3 | 50.4 | 69.1 |
| Ties-Merging | 59.8 | 58.6 | 70.7 | 79.7 | 86.2 | 72.1 | 98.3 | 54.2 | 72.4 |
| Consensus Merging | 65.7 | 63.6 | 76.5 | 77.2 | 81.7 | 70.3 | 97.0 | 57.1 | 73.6 |
| PCB Merging | 66.7 | 65.5 | 78.5 | 79.3 | 86.4 | 77.1 | 98.2 | 59.1 | 76.3 |
| **WUDI-Merging** (Ours) | **71.1** | **71.0** | **85.7** | **95.6** | **94.2** | **94.7** | **99.5** | **69.7** | **85.2**$_{\triangle 8.9}$ |

Table 2: Multi-task performance when merging ViT-L/14 models on 8-task vision benchmark.

| Method | SUN397 | Cars | RESISC45 | EuroSAT | SVHN | GTSRB | MNIST | DTD | Avg Acc |
|---|---|---|---|---|---|---|---|---|---|
| *Non-merging Methods* | | | | | | | | | |
| Pretrained | 66.8 | 77.7 | 71.0 | 59.9 | 58.4 | 50.5 | 76.3 | 55.3 | 64.5 |
| Individual | 82.3 | 92.4 | 97.4 | 100 | 98.1 | 99.2 | 99.7 | 84.1 | 94.2 |
| Traditional MTL | 80.8 | 90.6 | 96.3 | 96.3 | 97.6 | 99.1 | 99.6 | 84.4 | 93.5 |
| *Test-time Adaptation Methods* | | | | | | | | | |
| AdaMerging | 79.0 | 90.3 | 90.8 | 96.2 | 93.4 | 98.0 | 99.0 | 79.9 | 90.8 |
| AdaMerging++ | 79.4 | 90.3 | 91.6 | 97.4 | 93.4 | 97.5 | 99.0 | 79.2 | 91.0 |
| Representation Surgery | 75.7 | 84.4 | 93.1 | 98.8 | 91.3 | 93.4 | 99.1 | 76.1 | 89.0 |
| *Date-free Methods* | | | | | | | | | |
| Weight Averaging | 72.1 | 81.6 | 82.6 | 91.9 | 78.2 | 70.7 | 97.1 | 62.8 | 79.6 |
| Fisher Merging | 69.2 | 88.6 | 87.5 | 93.5 | 80.6 | 74.8 | 93.3 | 70.0 | 82.2 |
| RegMean | 73.3 | 81.8 | 86.1 | 97.0 | 88.0 | 84.2 | 98.5 | 60.8 | 83.7 |
| Task Arithmetic | 73.9 | 82.1 | 86.6 | 94.1 | 87.9 | 86.7 | 98.9 | 65.6 | 84.5 |
| Ties-Merging | 76.5 | 85.0 | 89.3 | 95.7 | 90.3 | 83.3 | 99.0 | 68.8 | 86.0 |
| Consensus Merging | 75.0 | 84.3 | 89.4 | 95.6 | 88.3 | 82.4 | 98.9 | 68.0 | 85.2 |
| PCB Merging | 76.8 | 86.2 | 89.4 | 96.5 | 88.3 | 91.0 | 98.6 | 73.6 | 87.5 |
| **WUDI-Merging** (Ours) | **81.0** | **91.0** | **94.2** | **99.2** | **96.3** | **98.1** | **99.6** | **81.2** | **92.6**$_{\triangle 5.1}$ |

our method lags behind supervised multi-task learning by only 4.7% and 0.9% on the ViT-B/32 and ViT-L/14 models. **Results of Language Tasks.** Table 5 summarizes the results for *discriminative language tasks*. In comparison to prior studies, WUDI-Merging demonstrates substantial performance improvements. Specifically, when evaluated against the advanced data-free approach Ties-Merging, WUDI-Merging achieved 19.7% improvement on the RoBERTa-Base model and 16.0% improvement on the RoBERTa-Large model. These findings strongly indicate the robust generalization capabilities of WUDI-Merging in language tasks. Table 3 presents the results for *generative language tasks*. We observed 4.3% improvement over Ties-Merging

(with DARE). However, regarding the results for code tasks, our performance remained at an average level without achieving a significant advantage. Our analysis suggests that this may be attributed to substantial conflicts between the math and code tasks, leading to a considerable decline in one task's performance when the other task excelled. This phenomenon is also reflected in the results from alternative methods. Nevertheless, our overall average score remains significantly ahead of the current state-of-the-art.

**Results of Merging LoRA Fine-tuned Models.** To further demonstrate the generalizability of our method on LoRA fine-tuned models, we supplemented the experiments on

Table 3: Performance of merging decoder-based WizardLM-13B (LM), WizardMath-13B (Math), and llama-2-13b-codealpaca (Code) on all the datasets, we reported average normalized score.

| Method | AlpacaEval | GSM8K | MATH | HumanEval | MBPP | Avg. |
|---|---|---|---|---|---|---|
| FT | 100.0 | 100.0 | 100.0 | 100.0 | 100.0 | 100.0 |
| Task Arithmetic | 102.7 | 91.0 | 70.5 | 50.0 | 87.7 | 80.4 |
| TIES-Merging | 98.1 | 97.4 | 68.1 | 60.0 | 89.4 | 82.6 |
| Task Arithmetic (w/ DARE) | 103.1 | 88.0 | 72.5 | 63.3 | **92.9** | 84.0 |
| TIES-Merging (w/ DARE) | 107.9 | 90.3 | 65.6 | **80.0** | 92.4 | 87.2 |
| **WUDI-Merging** (Ours) | **105.5** | **105.9** | **103.3** | 58.3 | 84.7 | **91.5**$_{\triangle 4.3}$ |

Table 4: Experimental results of merging Flan-T5-base (LoRA fine-tuned) models on all eight tasks.

| Method | CoLA | MNLI | MRPC | QNLI | QQP | RTE | SST2 | STSB | Avg. |
|---|---|---|---|---|---|---|---|---|---|
| Individual | 69.1 | 82.7 | 85.5 | 90.9 | 84.0 | 84.4 | 92.9 | 87.4 | 84.6 |
| Ties-Merging | 68.3 | 56.3 | 79.4 | 89.8 | 83.7 | 79.4 | 91.6 | 71.2 | 77.5 |
| AdaMerging++ | **69.1** | 60.3 | 78.4 | 90.0 | 83.6 | 79.1 | 91.6 | 74.1 | 78.3 |
| **WUDI-Merging** (Ours) | 68.6 | 79.0 | 77.7 | 87.2 | 83.1 | 75.8 | 93.2 | 85.0 | **81.2**$_{\triangle 2.9}$ |

Table 5: Multi-task performance when merging RoBERTa models on 8-task GLUE benchmark. Following Lu et al. (2024), we reported average normalized score.

| Method | RoBERTa-Base | RoBERTa-Large |
|---|---|---|
| Pretrained | 41.7 | 38.2 |
| Individual | 100.0 | 100.0 |
| Weight Averaging | 52.6 | 53.3 |
| Task Arithmetic | 67.8 | 70.9 |
| Ties-Merging | 64.7 | 72.4 |
| Task Arithmetic (w/ DARE) | 63.7 | 70.9 |
| Ties-Merging (w/ DARE) | 65.6 | 72.8 |
| **WUDI-Merging** (Ours) | **85.3**$_{\triangle 19.7}$ | **88.8**$_{\triangle 16.0}$ |

Table 6: Experimental results of merging Qwen-14B (LoRA fine-tuned) models on all four tasks.

| Method | MMLU | TruthfulQA | BBQ | CNN | Avg. |
|---|---|---|---|---|---|
| Individual | 68.35 | 53.34 | 93.53 | 19.46 | 58.67 |
| Task Arithmetic | 67.56 | 52.33 | 78.38 | 20.54 | 54.70 |
| Ties-Merging (w/ DARE) | 69.38 | 52.03 | 81.06 | 15.91 | 54.62 |
| **WUDI-Merging** (Ours) | 69.17 | 55.71 | 80.56 | 17.33 | **55.69**$_{\triangle 0.99}$ |

Flan-T5-base and Qwen-14. For merging LoRA, We first restore LoRA matrix back into the task vector ($\boldsymbol{\tau}_i = B_i A_i$), then apply WUDI-Merging directly to $\boldsymbol{\tau}_i$ to obtain $\boldsymbol{\tau}_m$, and merging it into $\boldsymbol{\theta}$. The experimental results obtained from merging Flan-T5-base (LoRA fine-tuned) models and Qwen-14B (LoRA fine-tuned) are shown in the Table 4 and Table 6. We observed 0.99% improvement over Ties-Merging (with DARE) in Qwen-14B models, and 2.99% improvement over AdaMerging++ in Flan-T5 models. Experimental results indicate that WUDI does not yield as large an improvement as observed in conventional settings. We hypothesize that

this arises from the random initialization of LoRA, which causes the task vector to accumulate gradients from nonzero and thereby introduces a small amount of noise. This noise degrades the task vector's ability to reconstruct the corresponding input, leading to the observed performance gap. Nevertheless, when evaluated under the LoRA-finetuned model, WUDI still attains SOTA performance.

### 4.3. Ablation Study

**Balanced Weight.** In Eq. 19, we introduce a dynamic weighting mechanism designed to balance the loss contributions across multiple experts. The experimental results presented in Table 7 demonstrate that the application of balanced weights significantly enhances generalization performance. This improvement further suggests that task vectors with larger magnitudes exhibit greater resilience to higher levels of interference.

**Stabilization Analysis on Solution Steps.** As shown in Fig. 4, we evaluate our method across a range of solution steps. The experimental results indicate a consistent increase in accuracy across all datasets and models over iterations, with convergence typically achieved within 100 to 200 iterations. These results underscore the generalization ability of our method and further highlight that only a limited number of steps are required to reach a solution, thereby demonstrating the efficiency of our approach.

**Stabilization Analysis on Task Numbers.** To further investigate the impact of task number on the performance of WUDI-Merging, we conducted experiments by varying the number of tasks. Following the methodology outlined in (Xiong et al., 2024; Wei et al., 2025), we sampled eight random subsets from the complete task set and calculated the average accuracy of the merged model across these sub-

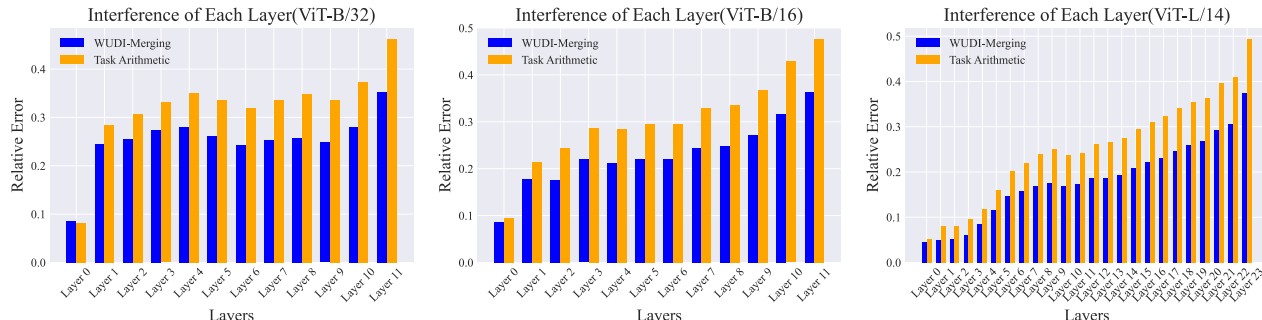

Figure 3: The interference of Task Arithmetic and WUDI-Merging for different layers.

Table 7: The impact of applying balanced weighting to the loss function. We report the performance when merging RoBERTa models on 8-task GLUE benchmark.

| Method | RoBERTa-Base | RoBERTa-Large |
|---|---|---|
| Task Arithmetic | 67.8 | 70.9 |
| Ours (w/o Balance) | 77.4 | 82.4 |
| Ours (w/ Balance) | **85.3** | **88.8** |

sets. As shown in Fig. 4, all methods exhibit a decline in average accuracy as the number of tasks increases, indicating heightened conflicts and interference among expert models. Baseline methods such as Task Arithmetic and Ties-Merging experience more pronounced performance drops, while WUDI-Merging consistently maintains higher accuracy as the number of tasks increases. This finding highlights the exceptional robustness of WUDI-Merging and its effective mitigation of task interference in multi-task learning scenarios.

### 4.4. Further Analysis

**Comparison between Adam Optimization and Closed-Form Solution.** While WUDI-Merging requires very few GPU resources for optimization, we also present a closed-form solution for scenarios where GPU resources are not available. Specifically, beginning from Eq. 19, we derive the closed-form expression outlined in Eq. 20, which we refer to as WUDI-Merging-CFS. Furthermore, we investigate the influence of the regularization term on the closed-form solution. As shown in Fig. 5, WUDI-Merging demonstrates greater stability by utilizing implicit optimization through the Adam optimizer. In contrast, WUDI-Merging-CFS is sensitive to the choice of regularization coefficient; moreover, its overall performance is typically inferior to that achieved with the Adam optimizer. Therefore, we suggest that employing gradient descent via Adam optimization is a more effective strategy for solving Eq. 19. This assertion is further corroborated by previous studies (Smith et al., 2021; Wang et al., 2022; Zou et al., 2021).

**Analysis on Rescaling Coefficient.** In this section, we

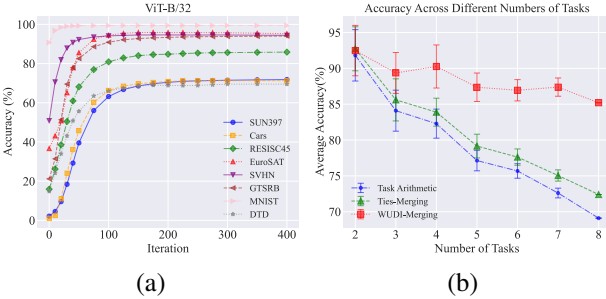

Figure 4: Figure (a) presents the results obtained at different stages of the solution process; Figure (b) presents the result of varying the number of tasks.

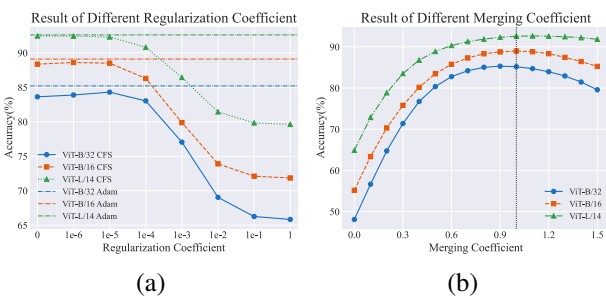

Figure 5: Figure (a) displays the results obtained using the closed-form solution with varying regularization coefficients; Figure (b) shows the results for different merging coefficients applied during the merging process.

evaluated WUDI-Merging's performance across a range of scaling coefficients to analyze the effect of the rescaling coefficient on WUDI-Merging. The rescaling coefficient is defined as $\boldsymbol{\theta_m} = \boldsymbol{\theta} + \epsilon \cdot \boldsymbol{\tau}_m$.

Where $\epsilon$ is the rescaling coefficient. As shown in Fig. 5, WUDI-Merging achieves optimal performance for the models ViT-B/32, ViT-B/16, and ViT-L/14 when $\epsilon$ is approximately 1. The results suggest that our method did not necessitate the tuning of the rescaling coefficient. In other words, WUDI-Merging is rescaling-free.

**The Interference of Each Layer.** The evaluation of layer-wise interference is presented in Fig. 3. The results indicate that interference accumulates progressively as the depth of

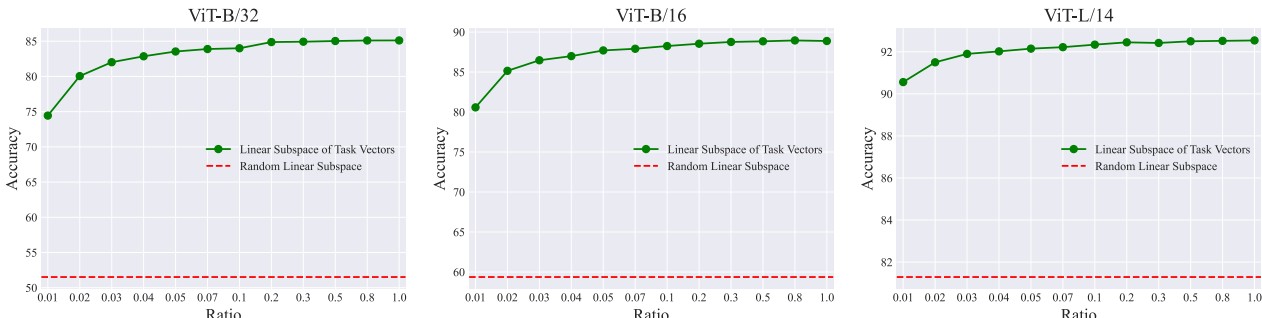

Figure 6: Performance comparison when utilizing a subset of task vectors or random vectors for optimizing the loss function.

Table 8: Computational time and gpu memory requirements.

| Model | Solving steps | Solving Time | GPU Memory |
|---|---|---|---|
| ViT-B/32 | 300 | 1min54s | 3.03GB |
| ViT-L/14 | 300 | 8min23s | 4.02GB |

Table 9: Comparison of different merging methods in terms of accuracy, time, and GPU memory usage under the setting of ViT-B/32.

| Method | Accuracy | Time | GPU Memory |
|---|---|---|---|
| Ties Merging | 72.4 | 4s | 0GB |
| Adamerging | 81.1 | 127min | 17.1GB |
| WUDI-Merging-CFS (CPU) | 84.4 | 5s | 0GB |
| WUDI-Merging-CFS (GPU) | 84.4 | 2s | 1.8GB |
| WUDI-Merging | **85.2** | 1min54s | 4.0GB |

the network increases. Specifically, task arithmetic introduces a relative error of nearly 50% in the final output. In contrast, our proposed method significantly mitigates this interference compared to task arithmetic and effectively slows the rate of interference accumulation.

**Selection of Linear Subspace.** Reconsidering Eq. 13, we investigate whether the entire task vector linear subspace is optimal for reconstruction in a data-free scenario. To verify this point, we employed both random vectors and a subset of the task vectors to optimize the loss. For the random vectors, we sample them from a Gaussian distribution where the mean and standard deviation are computed from the original task vectors:

$$\boldsymbol{\tau}_i^{\text{random}} \sim \mathcal{N}(\mu_i, \sigma_i^2) \qquad (22)$$

where $\mu_i = \text{mean}(\boldsymbol{\tau}_i)$, $\sigma_i = \text{std}(\boldsymbol{\tau}_i)$. The loss is given by:

$$\mathcal{L}_{\text{random}} = \sum \frac{1}{\|\boldsymbol{\tau}_i\|_F^2} \left(\boldsymbol{\tau}_m - \boldsymbol{\tau}_i\right) \left(\boldsymbol{\tau}_i^{\text{random}}\right)^\top \qquad (23)$$

For the subset of task vectors, we sample a random subvector from the original task vector as follows:

$$\boldsymbol{\tau}_i^{\text{sub}} = \boldsymbol{\tau}_i[\text{rand\_index}, \, :] \qquad (24)$$

The corresponding loss is computed as:

$$\mathcal{L}_{\text{sub}} = \sum_{i=1}^n \frac{1}{\|\boldsymbol{\tau}_i\|_F^2} \left(\boldsymbol{\tau}_m - \boldsymbol{\tau}_i\right) \left(\boldsymbol{\tau}_i^{\text{sub}}\right)^\top \qquad (25)$$

The experimental results are presented in Fig. 6. The findings indicate that the random linear subspace yielded poor performance. Conversely, we observed that utilizing a more complete task vector consistently led to improved results. Notably, when the reconstruction was performed using the linear subspace formed by the complete task vector, we achieved the best performance. These results underscore the necessity of employing the entire task vector for effective reconstruction.

**Computation Resource.** To assess the computational efficiency of our method, we measured the solving time and GPU memory usage on the ViT-B/32 and ViT-L/14 architectures. The number of solving steps was fixed at 300, and the Adam optimizer was employed with the loss optimized sequentially, layer by layer. As shown in Table 8, executing our method on the ViT-B/32 architecture requires approximately 2 minutes and 3 GB of GPU memory. Even for the larger ViT-L/14 architecture model, it only necessitates about 8 minutes and 4 GB of GPU memory. These results indicate that our method is highly computationally efficient.

## 5. Conclusion

In this paper, we demonstrate that the task vectors constitute an approximate linear subspace of the corresponding input for the linear layer. Therefore, we can minimize the interference among expert models guided by the task vector. Based on the insight, we propose WUDI-Merging, a simple yet effective model merging method that resolves interferences without requiring any additional data or rescaling coefficients. Extensive empirical evaluations demonstrate the effectiveness of our method. We believe that this work is one step toward a simple and general-purpose data-free model merging technique. Further research may consider a more detailed way to resolve the conflict based on our method, which has the potential to reach the performance of multitask learning in the data-free model merging paradigm.

## Acknowledgments

This work is supported by the National Key R&D Program of China (2022YFB4701400/4701402), SSTIC Grant (KJZD20230923115106012, KJZD20230923114916032, GJHZ20240218113604008), Beijing Key Lab of Networked Multimedia.

## Impact Statement

This paper presents work whose goal is to advance the field of Machine Learning. There are many potential societal consequences of our work, none which we feel must be specifically highlighted here.

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

# A. appendix

## A.1. Proof of theoretical results

### A.1.1. PROOF OF LEMMA 1

**Lemma** (Input consistency). *Let $x_l^p, x_l^q$ denote the input of the linear layer $l$ after $p$ and $q$ $(p > q)$ iterations during fine-tuning, respectively. Denote $\eta_i$ as the learning rate of iteration $i$. Assume that the model before layer $l$ is $\mathcal{C}_l$-Lipschitz continuous, and the gradient of loss is bounded in $\ell_2$-norm by $\mathcal{G}_l$, then:*

$$\|x_l^p - x_l^q\|_2 \leq \mathcal{C}_l \mathcal{G}_l \left( \sum_{i=q+1}^{p} \eta_i \right) \tag{26}$$

*Proof.* Let $f_{\sim l}(x; \boldsymbol{\theta}_{\sim l})$ denote the mapping from the input $x$ to the input of layer $l$, parameterized by $\boldsymbol{\theta}_{\sim l}$, which includes all parameters up to but not including layer $l$. Then,

$$x_l^t = f_{\sim l}(x; \boldsymbol{\theta}_{\sim l}^t) \tag{27}$$

The parameters are updated using gradient descent:

$$\boldsymbol{\theta}_{\sim l}^{i+1} = \boldsymbol{\theta}_{\sim l}^i - \eta_{i+1} \nabla_{\boldsymbol{\theta}_{\sim l}^i} \mathcal{L}(\boldsymbol{\theta}^i), \tag{28}$$

where $\eta_i$ is the learning rate at iteration $i$, and $\mathcal{L}(\boldsymbol{\theta}_{\sim l}^i)$ is the loss function evaluated at $\boldsymbol{\theta}^i$. From Eq. 28, the cumulative change in parameters is:

$$\boldsymbol{\theta}_{\sim l}^p = \boldsymbol{\theta}_{\sim l}^0 - \sum_{i=1}^{p} \eta_i \nabla_{\boldsymbol{\theta}_{\sim l}^{i-1}} \mathcal{L}(\boldsymbol{\theta}^{i-1}), \quad \boldsymbol{\theta}_{\sim l}^q = \boldsymbol{\theta}_{\sim l}^0 - \sum_{i=1}^{q} \eta_i \nabla_{\boldsymbol{\theta}_{\sim l}^{i-1}} \mathcal{L}(\boldsymbol{\theta}^{i-1}) \tag{29}$$

The difference between $x_l^p$ and $x_l^p$ can be formulated as:

$$x_l^p - x_l^q = f_{\sim l}(x; \boldsymbol{\theta}_{\sim l}^p) - f_{\sim l}(x; \boldsymbol{\theta}_{\sim l}^q) \tag{30}$$

To analyze the magnitude of the change, we consider the $\ell_2$-norm of the output difference.:

$$\|x_l^p - x_l^q\|_2 = \|f_{\sim l}(x; \boldsymbol{\theta}_{\sim l}^p) - f_{\sim l}(x; \boldsymbol{\theta}_{\sim l}^q)\|_2 \tag{31}$$

Assume that the model before layer $l$ is $\mathcal{C}_l$-Lipschitz continuous, and the gradient of loss is bounded in $\ell_2$-norm by $\mathcal{G}_l$ then:

$$\|x_l^p - x_l^q\|_2 = \|f_{\sim l}(x; \boldsymbol{\theta}_{\sim l}^p) - f_{\sim l}(x; \boldsymbol{\theta}_{\sim l}^q)\|_2 \tag{32}$$

$$\leq \mathcal{C}_l \|\boldsymbol{\theta}_{\sim l}^p - \boldsymbol{\theta}_{\sim l}^q\|_2 \tag{33}$$

$$= \mathcal{C}_l \|(\boldsymbol{\theta}_{\sim l}^0 - \sum_{i=1}^{p} \eta_i \nabla_{\boldsymbol{\theta}_{\sim l}^{i-1}} \mathcal{L}(\boldsymbol{\theta}^{i-1})) - (\boldsymbol{\theta}_{\sim l}^0 - \sum_{i=1}^{q} \eta_i \nabla_{\boldsymbol{\theta}_{\sim l}^{i-1}} \mathcal{L}(\boldsymbol{\theta}^{i-1}))\|_2 \tag{34}$$

$$= \mathcal{C}_l \|\sum_{i=q+1}^{p} \eta_i \nabla_{\boldsymbol{\theta}_{\sim l}^{i-1}} \mathcal{L}(\boldsymbol{\theta}^{i-1})\|_2 \tag{35}$$

$$= \mathcal{C}_l \sum_{i=q+1}^{p} \eta_i \|\nabla_{\boldsymbol{\theta}_{\sim l}^{i-1}} \mathcal{L}(\boldsymbol{\theta}^{i-1})\|_2 \tag{36}$$

$$\leq \mathcal{C}_l \mathcal{G}_l \left( \sum_{i=q+1}^{p} \eta_i \right) \tag{37}$$

where the Eq.33 follows from the $\mathcal{C}_l$-Lipschitz continuity of $f_{\sim l}$, and Eq. 37 uses the assumption that the gradient is bounded in $\ell_2$-norm by $\mathcal{G}_l$.

∎

A.1.2. PROOF OF PROPOSITION 1

**Proposition** (Approximate Linear Combination). *Let $\boldsymbol{\tau}_l^k$ denote the task vector of neuron $k$ in linear layer $l$. Consider $N$ input samples $\{\boldsymbol{x}_n\}_{n=1}^N$, and let $\boldsymbol{x}_{n,l}^t$ denote the corresponding input to layer $l$ after $t$ iteration for sample $\boldsymbol{x}_n$ during fine-tuning. Assume the gradient of the loss with respect to the product $\boldsymbol{\theta}_{l,k}^{t-1}\boldsymbol{x}_{n,l}^t$ is bounded by $\Gamma_{l,k}$. Then, the following inequality holds:*

$$\left\| \boldsymbol{\tau}_l^k - \sum_{n=1}^N \beta_{n,l}^k (\boldsymbol{x}_{n,l}^{\mathrm{T}})^\top \right\|_2 \leq \Phi_{l,k} \cdot \left( \sum_{t=1}^{\mathrm{T}} \sum_{i=t}^{\mathrm{T}} \eta_t \eta_i \right). \tag{38}$$

*where $\Phi_{l,k} = N \cdot \mathcal{C}_l \cdot \mathcal{G}_l \cdot \Gamma_{l,k}$ and $\beta_{n,l}^k = \sum_{t=1}^{\mathrm{T}} -\eta_t \frac{\partial \mathcal{L}(\boldsymbol{\theta}^t)}{\partial(\boldsymbol{\theta}_{l,k}^{t-1}\boldsymbol{x}_{n,l}^t)}$.*

*Proof.* The task vector is calculated as the accumulation of gradients:

$$\boldsymbol{\tau}_l^k = \sum_{t=1}^{\mathrm{T}} -\eta_t \cdot \frac{\partial \mathcal{L}(\boldsymbol{\theta}^{t-1})}{\partial \boldsymbol{\theta}_{l,k}^{t-1}} \tag{39}$$

$$= \sum_{t=1}^{\mathrm{T}} -\eta_t \sum_{n=1}^N \frac{\partial \mathcal{L}(\boldsymbol{\theta}^{t-1})}{\partial(\boldsymbol{\theta}_{l,k}^{t-1}\boldsymbol{x}_{n,l}^{t-1})} \cdot \frac{\partial(\boldsymbol{\theta}_{l,k}^{t-1}\boldsymbol{x}_{n,l}^{t-1})}{\partial \boldsymbol{\theta}_{l,k}^{t-1}} \tag{40}$$

$$= \sum_{t=1}^{\mathrm{T}} -\eta_t \sum_{n=1}^N \frac{\partial \mathcal{L}(\boldsymbol{\theta}^{t-1})}{\partial(\boldsymbol{\theta}_{l,k}^{t-1}\boldsymbol{x}_{n,l}^{t-1})} \cdot (\boldsymbol{x}_{n,l}^{t-1})^\top \tag{41}$$

Let $\gamma_{n,l,k}^t$ denote the gradient of the loss with respect to the product $\boldsymbol{\theta}_{l,k}^{t-1}\boldsymbol{x}_{n,l}^{t-1}$, that is, $\gamma_{n,l,k}^t = \frac{\partial \mathcal{L}(\boldsymbol{\theta}^{t-1})}{\partial(\boldsymbol{\theta}_{l,k}^{t-1}\boldsymbol{x}_{n,l}^{t-1})}$, then:

$$\boldsymbol{\tau}_l^k = \sum_{t=1}^{\mathrm{T}} -\eta_t \sum_{n=1}^N \gamma_{n,l,k}^t \cdot (\boldsymbol{x}_{n,l}^{t-1})^\top \tag{42}$$

$$= \sum_{t=1}^{\mathrm{T}} -\eta_t \sum_{n=1}^N \gamma_{n,l,k}^t \cdot ((\boldsymbol{x}_{n,l}^{t-1})^\top - (\boldsymbol{x}_{n,l}^{\mathrm{T}})^\top + (\boldsymbol{x}_{n,l}^{\mathrm{T}})^\top) \tag{43}$$

Rearranging, we obtain:

$$\boldsymbol{\tau}_l^k - \sum_{t=1}^{\mathrm{T}} -\eta_t \sum_{n=1}^N \gamma_{n,l,k}^t \cdot (\boldsymbol{x}_{n,l}^{\mathrm{T}})^\top = \sum_{t=1}^{\mathrm{T}} -\eta_t \sum_{n=1}^N \gamma_{n,l,k}^t \cdot ((\boldsymbol{x}_{n,l}^{t-1})^\top - (\boldsymbol{x}_{n,l}^{\mathrm{T}})^\top) \tag{44}$$

Taking the $\ell_2$-norm of both sides, we have:

$$\left\| \boldsymbol{\tau}_l^k - \sum_{t=1}^{\mathrm{T}} -\eta_t \sum_{n=1}^N \gamma_{n,l,k}^t \cdot (\boldsymbol{x}_{n,l}^{\mathrm{T}})^\top \right\|_2 = \left\| \sum_{t=1}^{\mathrm{T}} -\eta_t \sum_{n=1}^N \gamma_{n,l,k}^t \cdot ((\boldsymbol{x}_{n,l}^{t-1})^\top - (\boldsymbol{x}_{n,l}^{\mathrm{T}})^\top) \right\|_2 \tag{45}$$

$$\leq \sum_{t=1}^{\mathrm{T}} \eta_t \sum_{n=1}^N |\gamma_{n,l,k}^t| \cdot \left\| (\boldsymbol{x}_{n,l}^{t-1})^\top - (\boldsymbol{x}_{n,l}^{\mathrm{T}})^\top \right\|_2 \tag{46}$$

Assume $\gamma_{n,l,k}^t = \frac{\partial \mathcal{L},(\boldsymbol{\theta}^{t-1})}{\partial(\boldsymbol{\theta}_{l,k}^{t-1}\boldsymbol{x}_{n,l}^{t-1})}$ is bounded by $\Gamma_{l,k}$, then:

$$\left\| \boldsymbol{\tau}_l^k - \sum_{t=1}^{\mathrm{T}} -\eta_t \sum_{n=1}^N \gamma_{n,l,k}^t \cdot (\boldsymbol{x}_{n,l}^{\mathrm{T}})^\top \right\|_2 \leq \sum_{t=1}^{\mathrm{T}} \eta_t \sum_{n=1}^N \Gamma_{l,k} \cdot \left\| (\boldsymbol{x}_{n,l}^{t-1})^\top - (\boldsymbol{x}_{n,l}^{\mathrm{T}})^\top \right\|_2 \tag{47}$$

Applying Lemma.1 to Eq.47, then:

$$\left\| \boldsymbol{\tau}_l^k - \sum_{t=1}^{\mathrm{T}} -\eta_t \sum_{n=1}^{N} \gamma_{n,l,k}^t \cdot (\boldsymbol{x}_{n,l}^{\mathrm{T}})^\top \right\|_2 \leq \sum_{t=1}^{\mathrm{T}} \eta_t \sum_{n=1}^{N} \Gamma_{l,k} \cdot \mathcal{C}_l \mathcal{G}_l (\sum_{i=t}^{\mathrm{T}} \eta_i) \tag{48}$$

$$= N \cdot \Gamma_{l,k} \cdot \mathcal{C}_l \cdot \mathcal{G}_l \cdot \sum_{t=1}^{\mathrm{T}} \eta_t (\sum_{i=t}^{\mathrm{T}} \eta_i) \tag{49}$$

$$= N \cdot \Gamma_{l,k} \cdot \mathcal{C}_l \cdot \mathcal{G}_l \cdot (\sum_{t=1}^{\mathrm{T}} \sum_{i=t}^{\mathrm{T}} \eta_t \eta_i) \tag{50}$$

Denote $\beta_{n,l}^k = \sum_{t=1}^{\mathrm{T}} -\eta_t \gamma_{n,l,k}^t$ and $\Phi_{l,k} = N \cdot \mathcal{C}_l \cdot \mathcal{G}_l \cdot \Gamma_{l,k}$, then we can reformulate Eq.50 as follows:

$$\left\| \boldsymbol{\tau}_l^k - \sum_{n=1}^{N} \beta_{n,l}^k (\boldsymbol{x}_{n,l}^{\mathrm{T}})^\top \right\|_2 \leq \Phi_{l,k} \cdot (\sum_{t=1}^{\mathrm{T}} \sum_{i=t}^{\mathrm{T}} \eta_t \eta_i) \tag{51}$$

∎

### A.1.3. PROOF OF THEOREM 1

**Theorem** (Upper bound of interference). *Denote $\boldsymbol{x}_{i,l}$ as the linear layer $l$'s input of task $i$, $\boldsymbol{\delta}_{i,l}$ and $\boldsymbol{\tau}_{i,l}$ as the linear layer $l$'s interference vector and task vector of task $i$, respectively, then:*

$$\mathbb{E}_{\boldsymbol{x}_{i,l}\sim p(\boldsymbol{x}_{i,l})}\|\boldsymbol{\delta}_{i,l}\boldsymbol{x}_{i,l}\|_2^2 \leq \omega_{i,l}^1 \cdot \left\|\boldsymbol{\delta}_i(\boldsymbol{\tau}_{i,l})^\top\right\|_F^2 + \omega_{i,l}^2 \cdot \|\boldsymbol{\delta}_i\|_F^2 \tag{52}$$

*Where $\omega_{i,l}^1$ and $\omega_{i,l}^2$ is the reconstruction constant.*

*Proof.* According to Lemma 2, each neuron of the task vector associated with a linear layer can be interpreted as a weighted sum of the corresponding input samples. Consequently, the entire task vector forms an approximate linear subspace of the input space, which motivates us to reconstruct the input using the task vector:

$$\boldsymbol{x}_{i,l} = \sum_{k=1}^{K} \alpha_{i,l}^k(\boldsymbol{x}_{i,l})(\boldsymbol{\tau}_{i,l}^k)^\top + \varepsilon(\boldsymbol{x}_{i,l}), \quad \boldsymbol{x}_{i,l} \in \mathcal{D}_{i,l} \tag{53}$$

where $\mathcal{D}_{i,l}$ is the input domain of layer $l$ for task $i$, $\alpha_{i,l}^k(\boldsymbol{x}_{i,l})$ are the reconstruction coefficients of $\boldsymbol{x}_{i,l}$, $(\boldsymbol{\tau}_{i,l}^k)^\top$ are $k$-th neuron of task vector $\boldsymbol{\tau}_{i,l}$, and $\varepsilon(\boldsymbol{x}_{i,l})$ represents the reconstruction error. The expectation of interference is formulated as follows:

$$\mathbb{E}_{\boldsymbol{x}_{i,l}\sim p(\boldsymbol{x}_{i,l})}\|\boldsymbol{\delta}_{i,l}\boldsymbol{x}_{i,l}\|_2^2 = \int_{\boldsymbol{x}_{i,l}\in\mathcal{D}_{i,l}} \|\boldsymbol{\delta}_{i,l}\boldsymbol{x}_{i,l}\|_2^2 \cdot p(\boldsymbol{x}_{i,l})\, d\boldsymbol{x}_{i,l}. \tag{54}$$

Reconstructing $\boldsymbol{x}_{i,l}$ by Eq.53, then:

$$\mathbb{E}_{\boldsymbol{x}_{i,l}\sim p(\boldsymbol{x}_{i,l})}\|\boldsymbol{\delta}_{i,l}\boldsymbol{x}_{i,l}\|_2^2 = \int_{\boldsymbol{x}_{i,l}\in\mathcal{D}_{i,l}} \left\|\boldsymbol{\delta}_i\left(\sum_{k=1}^{K}\alpha_{i,l}^k(\boldsymbol{x}_{i,l})(\boldsymbol{\tau}_{i,l}^k)^\top + \varepsilon(\boldsymbol{x}_{i,l})\right)\right\|_2^2 p(\boldsymbol{x}_{i,l})\, d\boldsymbol{x}_{i,l}. \tag{55}$$

Scaling Eq.55 by the triangle inequality, then:

$$\mathbb{E}_{\boldsymbol{x}_{i,l}\sim p(\boldsymbol{x}_{i,l})}\|\boldsymbol{\delta}_{i,l}\boldsymbol{x}_{i,l}\|_2^2 = \int_{\boldsymbol{x}_{i,l}\in\mathcal{D}_{i,l}} \left\|\boldsymbol{\delta}_i\left(\sum_{k=1}^{K}\alpha_{i,l}^k(\boldsymbol{x}_{i,l})(\boldsymbol{\tau}_{i,l}^k)^\top + \varepsilon(\boldsymbol{x}_{i,l})\right)\right\|_2^2 p(\boldsymbol{x}_{i,l})\, d\boldsymbol{x}_{i,l}. \tag{56}$$

$$= \int_{\boldsymbol{x}_{i,l}\in\mathcal{D}_{i,l}} \left\|\sum_{k=1}^{K}\alpha_{i,l}^k(\boldsymbol{x}_{i,l})\boldsymbol{\delta}_i(\boldsymbol{\tau}_{i,l}^k)^\top + \boldsymbol{\delta}_i\varepsilon(\boldsymbol{x}_{i,l})\right\|_2^2 p(\boldsymbol{x}_{i,l})\, d\boldsymbol{x}_{i,l}. \tag{57}$$

$$= \int_{\boldsymbol{x}_{i,l}\in\mathcal{D}_{i,l}} \left(\left\|\sum_{k=1}^{K}\alpha_{i,l}^k(\boldsymbol{x}_{i,l})\boldsymbol{\delta}_i(\boldsymbol{\tau}_{i,l}^k)^\top + \boldsymbol{\delta}_i\varepsilon(\boldsymbol{x}_{i,l})\right\|_2\right)^2 p(\boldsymbol{x}_{i,l})\, d\boldsymbol{x}_{i,l}. \tag{58}$$

$$\leq \int_{\boldsymbol{x}_{i,l}\in\mathcal{D}_{i,l}} \left(\sum_{k=1}^{K}|\alpha_{i,l}^k(\boldsymbol{x}_{i,l})|\left\|\boldsymbol{\delta}_i(\boldsymbol{\tau}_{i,l}^k)^\top\right\|_2 + \|\boldsymbol{\delta}_i\varepsilon(\boldsymbol{x}_{i,l})\|_2\right)^2 p(\boldsymbol{x}_{i,l})\, d\boldsymbol{x}_{i,l}. \tag{59}$$

Applying the Cauchy-Schwarz inequality the part in brackets, then we can get:

$$\left(\sum_{k=1}^{K}|\alpha_{i,l}^k(\boldsymbol{x}_{i,l})|\left\|\boldsymbol{\delta}_i(\boldsymbol{\tau}_{i,l}^k)^\top\right\|_2 + \|\boldsymbol{\delta}_i\varepsilon(\boldsymbol{x}_{i,l})\|_2\right)^2 \tag{60}$$

$$= \left(\sum_{k=1}^{K}|\alpha_{i,l}^k(\boldsymbol{x}_{i,l})|\cdot\left\|\boldsymbol{\delta}_i(\boldsymbol{\tau}_{i,l}^k)^\top\right\|_2 + 1\cdot\|\boldsymbol{\delta}_i\varepsilon(\boldsymbol{x}_{i,l})\|_2\right)^2 \tag{61}$$

$$\leq \left(\sum_{k=1}^{K}|\alpha_{i,l}^k(\boldsymbol{x}_{i,l})|^2 + 1^2\right)\left(\sum_{k=1}^{K}\left\|\boldsymbol{\delta}_i(\boldsymbol{\tau}_{i,l}^k)^\top\right\|_2^2 + \|\boldsymbol{\delta}_i\varepsilon(\boldsymbol{x}_{i,l})\|_2^2\right) \tag{62}$$

Applying Eq.62 to Eq.59, then:

$$\mathbb{E}_{\boldsymbol{x}_{i,l}\sim p(\boldsymbol{x}_{i,l})}\|\boldsymbol{\delta}_{i,l}\boldsymbol{x}_{i,l}\|_2^2 \leq \int_{\boldsymbol{x}_{i,l}\in\mathcal{D}_{i,l}} \left(\sum_{k=1}^{K}|\alpha_{i,l}^k(\boldsymbol{x}_{i,l})|\,\|\boldsymbol{\delta}_i(\boldsymbol{\tau}_{i,l}^k)^\top\|_2 + \|\boldsymbol{\delta}_i\boldsymbol{\varepsilon}(\boldsymbol{x}_{i,l})\|_2\right)^2 p(\boldsymbol{x}_{i,l})\,d\boldsymbol{x}_{i,l}. \tag{63}$$

$$\leq \int_{\boldsymbol{x}_{i,l}\in\mathcal{D}_{i,l}} \left(\sum_{k=1}^{K}|\alpha_{i,l}^k(\boldsymbol{x}_{i,l})|^2 + 1^2\right)\left(\sum_{k=1}^{K}\|\boldsymbol{\delta}_i(\boldsymbol{\tau}_{i,l}^k)^\top\|_2^2 + \|\boldsymbol{\delta}_i\boldsymbol{\varepsilon}(\boldsymbol{x}_{i,l})\|_2^2\right) p(\boldsymbol{x}_{i,l})\,d\boldsymbol{x}_{i,l}. \tag{64}$$

$$\leq \int_{\boldsymbol{x}_{i,l}\in\mathcal{D}_{i,l}} \left(\sum_{k=1}^{K}|\alpha_{i,l}^k(\boldsymbol{x}_{i,l})|^2 + 1^2\right)\left(\sum_{k=1}^{K}\|\boldsymbol{\delta}_i(\boldsymbol{\tau}_{i,l}^k)^\top\|_2^2 + \|\boldsymbol{\delta}_i\|_F^2\,\|\boldsymbol{\varepsilon}(\boldsymbol{x}_{i,l})\|_2^2\right) p(\boldsymbol{x}_{i,l})\,d\boldsymbol{x}_{i,l}. \tag{65}$$

Consider that $\sum_{k=1}^{K}\left\|\boldsymbol{\delta}_i(\boldsymbol{\tau}_{i,l}^k)^\top\right\|_2^2 = \left\|\boldsymbol{\delta}_i(\boldsymbol{\tau}_{i,l})^\top\right\|_F^2$ and let $\boldsymbol{\alpha}_{i,l}(\boldsymbol{x}_{i,l})$ denote $\left(\sum_{k=1}^{K}\left|\alpha_{i,l}^k(\boldsymbol{x}_{i,l})\right|^2 + 1^2\right)$, then we can reformulated Eq.65 as follows:

$$\mathbb{E}_{\boldsymbol{x}_{i,l}\sim p(\boldsymbol{x}_{i,l})}\|\boldsymbol{\delta}_{i,l}\boldsymbol{x}_{i,l}\|_2^2 \leq \int_{\boldsymbol{x}_{i,l}\in\mathcal{D}_{i,l}} \boldsymbol{\alpha}_{i,l}(\boldsymbol{x}_{i,l})\left(\left\|\boldsymbol{\delta}_i(\boldsymbol{\tau}_{i,l})^\top\right\|_F^2 + \|\boldsymbol{\delta}_i\|_F^2\,\|\boldsymbol{\varepsilon}(\boldsymbol{x}_{i,l})\|_2^2\right) p(\boldsymbol{x}_{i,l})\,d\boldsymbol{x}_{i,l} \tag{66}$$

$$= \left(\int_{\boldsymbol{x}_{i,l}\in\mathcal{D}_{i,l}} \boldsymbol{\alpha}_{i,l}(\boldsymbol{x}_{i,l})p(\boldsymbol{x}_{i,l})\,d\boldsymbol{x}_{i,l}\right)\left\|\boldsymbol{\delta}_i(\boldsymbol{\tau}_{i,l})^\top\right\|_F^2 \tag{67}$$

$$+ \left(\int_{\boldsymbol{x}_{i,l}\in\mathcal{D}_{i,l}} \boldsymbol{\alpha}_{i,l}(\boldsymbol{x}_{i,l})\,\|\boldsymbol{\varepsilon}(\boldsymbol{x}_{i,l})\|_2^2\,p(\boldsymbol{x}_{i,l})\,d\boldsymbol{x}_{i,l}\right)\|\boldsymbol{\delta}_i\|_F^2 \tag{68}$$

$$= \left(\mathbb{E}_{\boldsymbol{x}_{i,l}\sim p(\boldsymbol{x}_{i,l})}\boldsymbol{\alpha}_{i,l}(\boldsymbol{x}_{i,l})\right)\cdot\left\|\boldsymbol{\delta}_i(\boldsymbol{\tau}_{i,l})^\top\right\|_F^2 + \left(\mathbb{E}_{\boldsymbol{x}_{i,l}\sim p(\boldsymbol{x}_{i,l})}\boldsymbol{\alpha}_{i,l}(\boldsymbol{x}_{i,l})\,\|\boldsymbol{\varepsilon}(\boldsymbol{x}_{i,l})\|_2^2\right)\cdot\|\boldsymbol{\delta}_i\|_F^2 \tag{69}$$

Considering that when $\boldsymbol{\tau}_{i,l}$ and $\boldsymbol{x}_{i,l}$ are fixed, the reconstruction error and reconstruction coefficient are fixed, therefore $\mathbb{E}_{\boldsymbol{x}_{i,l}\sim p(\boldsymbol{x}_{i,l})}\boldsymbol{\alpha}_{i,l}(\boldsymbol{x}_{i,l})$ and $\mathbb{E}_{\boldsymbol{x}_{i,l}\sim p(\boldsymbol{x}_{i,l})}\boldsymbol{\alpha}_{i,l}(\boldsymbol{x}_{i,l})\,\|\boldsymbol{\varepsilon}(\boldsymbol{x}_{i,l})\|_2^2$ is a constant. Denote $\mathbb{E}_{\boldsymbol{x}_{i,l}\sim p(\boldsymbol{x}_{i,l})}\boldsymbol{\alpha}_{i,l}(\boldsymbol{x}_{i,l})$ and $\mathbb{E}_{\boldsymbol{x}_{i,l}\sim p(\boldsymbol{x}_{i,l})}\boldsymbol{\alpha}_{i,l}(\boldsymbol{x}_{i,l})\,\|\boldsymbol{\varepsilon}(\boldsymbol{x}_{i,l})\|_2^2$ as $\omega_{i,l}^1$ and $\omega_{i,l}^2$ respectively. Therefore we can get:

$$\mathbb{E}_{\boldsymbol{x}_{i,l}\sim p(\boldsymbol{x}_{i,l})}\|\boldsymbol{\delta}_{i,l}\boldsymbol{x}_{i,l}\|_2^2 \leq \omega_{i,l}^1\cdot\left\|\boldsymbol{\delta}_i(\boldsymbol{\tau}_{i,l})^\top\right\|_F^2 + \omega_{i,l}^2\cdot\|\boldsymbol{\delta}_i\|_F^2 \tag{70}$$

$\blacksquare$

A.1.4. THE DETAILED DERIVATION FOR THE CLOSE-FORM SOLUTION.

**Considering the objective with the regularization term**. The objective can be formulated as follows:

$$\min_{\boldsymbol{\tau}_{m,l}} obj = \sum_i \frac{1}{\|\boldsymbol{\tau}_{i,l}\|_F^2} \left( \left\| (\boldsymbol{\tau}_{m,l} - \boldsymbol{\tau}_{i,l}) \boldsymbol{\tau}_{i,l}^\top \right\|_F^2 + \omega \cdot \|\boldsymbol{\tau}_{m,l} - \boldsymbol{\tau}_{i,l}\|_F^2 \right) \tag{71}$$

$$\Leftrightarrow \min_{\boldsymbol{\tau}_{m,l}} \sum_i \frac{1}{\|\boldsymbol{\tau}_{i,l}\|_F^2} \left( \left\| \boldsymbol{\tau}_{m,l} \boldsymbol{\tau}_{i,l}^\top - \boldsymbol{\tau}_{i,l} \boldsymbol{\tau}_{i,l}^\top \right\|_F^2 + \omega \cdot \|\boldsymbol{\tau}_{m,l} - \boldsymbol{\tau}_{i,l}\|_F^2 \right) \tag{72}$$

Calculate the gradient of $\left\| \boldsymbol{\tau}_{m,l} \boldsymbol{\tau}_{i,l}^\top - \boldsymbol{\tau}_{i,l} \boldsymbol{\tau}_{i,l}^\top \right\|_F^2$:

$$\frac{\partial}{\partial \boldsymbol{\tau}_{m,l}} \left\| \boldsymbol{\tau}_{m,l} \boldsymbol{\tau}_{i,l}^\top - \boldsymbol{\tau}_{i,l} \boldsymbol{\tau}_{i,l}^\top \right\|_F^2 = 2(\boldsymbol{\tau}_{m,l} \boldsymbol{\tau}_{i,l}^\top - \boldsymbol{\tau}_{i,l} \boldsymbol{\tau}_{i,l}^\top) \boldsymbol{\tau}_{i,l} \tag{73}$$

Calculate the gradient of $\|\boldsymbol{\tau}_{m,l} - \boldsymbol{\tau}_{i,l}\|_F^2$

$$\frac{\partial}{\partial \boldsymbol{\tau}_{m,l}} \|\boldsymbol{\tau}_{m,l} - \boldsymbol{\tau}_{i,l}\|_F^2 = 2(\boldsymbol{\tau}_{m,l} - \boldsymbol{\tau}_{i,l}) \tag{74}$$

Combine the gradients:

$$\nabla_{\boldsymbol{\tau}_{m,l}} obj = \sum_i \frac{1}{\|\boldsymbol{\tau}_{i,l}\|_F^2} \left( 2(\boldsymbol{\tau}_{m,l} \boldsymbol{\tau}_{i,l}^\top - \boldsymbol{\tau}_{i,l} \boldsymbol{\tau}_{i,l}^\top) \boldsymbol{\tau}_{i,l} + 2\omega(\boldsymbol{\tau}_{m,l} - \boldsymbol{\tau}_{i,l}) \right) \tag{75}$$

To find the minimizer $\boldsymbol{\tau}_{m,l}$, set $\nabla_{\boldsymbol{\tau}_{m,l}} obj = 0$:

$$\sum_i \frac{1}{\|\boldsymbol{\tau}_{i,l}\|_F^2} \left( (\boldsymbol{\tau}_{m,l} \boldsymbol{\tau}_{i,l}^\top - \boldsymbol{\tau}_{i,l} \boldsymbol{\tau}_{i,l}^\top) \boldsymbol{\tau}_{i,l} + \omega(\boldsymbol{\tau}_{m,l} - \boldsymbol{\tau}_{i,l}) \right) = 0 \tag{76}$$

The equation can be written in the form:

$$\boldsymbol{\tau}_{m,l} A = B \tag{77}$$

where:

$$A = \sum_i \frac{1}{\|\boldsymbol{\tau}_{i,l}\|_F^2} \left( \boldsymbol{\tau}_{i,l}^\top \boldsymbol{\tau}_{i,l} + \omega I \right) \tag{78}$$

$$B = \sum_i \frac{1}{\|\boldsymbol{\tau}_{i,l}\|_F^2} \left( \boldsymbol{\tau}_{i,l} \boldsymbol{\tau}_{i,l}^\top \boldsymbol{\tau}_{i,l} + \omega \boldsymbol{\tau}_{i,l} \right) \tag{79}$$

Assuming $A$ is invertible, the optimal $\boldsymbol{\tau}_{m,l}$ is:

$$\boldsymbol{\tau}_{m,l} = B A^{-1} \tag{80}$$

Substituting back $A$ and $B$:

$$\boldsymbol{\tau}_{m,l} = \left( \sum_i \frac{1}{\|\boldsymbol{\tau}_{i,l}\|_F^2} (\boldsymbol{\tau}_{i,l} \boldsymbol{\tau}_{i,l}^\top \boldsymbol{\tau}_{i,l} + \omega \boldsymbol{\tau}_{i,l}) \right) \left( \sum_i \frac{1}{\|\boldsymbol{\tau}_{i,l}\|_F^2} (\boldsymbol{\tau}_{i,l}^\top \boldsymbol{\tau}_{i,l} + \omega I) \right)^{-1} \tag{81}$$

$$= \left( \sum_i \frac{1}{\|\boldsymbol{\tau}_{i,l}\|_F^2} (\boldsymbol{\tau}_{i,l} (\boldsymbol{\tau}_{i,l}^\top \boldsymbol{\tau}_{i,l} + \omega I)) \right) \left( \sum_i \frac{1}{\|\boldsymbol{\tau}_{i,l}\|_F^2} (\boldsymbol{\tau}_{i,l}^\top \boldsymbol{\tau}_{i,l} + \omega I)) \right)^{-1} \tag{82}$$

## A.2. Towards understanding the conflict in Model Merging

According to 3.2, the conflict among different experts arises from the interaction between the interference vector and the input. Moreover, this conflict may be exacerbated by the overlap of input domains. This phenomenon is further evidenced by the results of the vision and language tasks. Specifically, the domain overlap among various visual tasks is limited, whereas the inputs across different language tasks exhibit a larger overlap. Consequently, the current model-merging approach allows for a high degree of recovery in vision tasks, while there remains a notable disparity between the performance of the merged model and that of individual experts in language tasks. To address this limitation, it may be necessary to adopt a data processing perspective aimed at reducing the overlap of input domains across different language tasks.

## A.3. The impact of linear layer

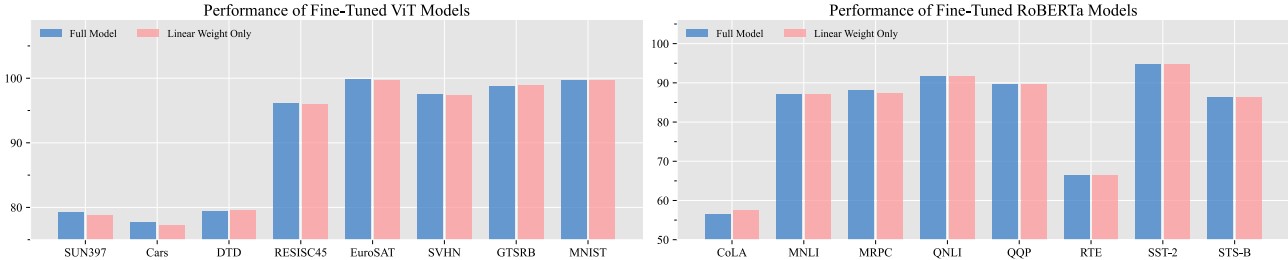

Figure 7: The impact of linear layer

Motivated by (Jin et al., 2023), we only apply our method to the linear layer in the model. To further demonstrate the effectiveness, we only apply the task vector of the linear layer to the pretrained model. The result is shown in Fig.7. The result shows that only using the task vector of the linear keeps most of the ability of the model. This means focusing on the linear layer is the key to the model merging problems.

### A.4. Additional Experiments

In this subsection, we provide the result of ViT-B/16 and the detailed result of *discriminative language tasks* in the following table, which further demonstrates the generalization of our method.

Table 10: Multi-task performance when merging ViT-B/16 models on eight tasks.

| Method | SUN397 | Cars | RESISC45 | EuroSAT | SVHN | GTSRB | MNIST | DTD | Avg Acc |
|---|---|---|---|---|---|---|---|---|---|
| *Non-merging Methods* | | | | | | | | | |
| Pretrained | 63.8 | 64.6 | 65.7 | 54.5 | 52.0 | 43.3 | 51.7 | 45.1 | 55.0 |
| Individual | 81.8 | 86.8 | 96.9 | 99.7 | 97.8 | 99.1 | 99.7 | 82.0 | 92.9 |
| *Test-time Adaption Methods* | | | | | | | | | |
| TW AdaMerging | 64.4 | 64.2 | 75.4 | 86.7 | 86.3 | 86.7 | 97.6 | 46.9 | 76.0 |
| LW AdaMerging | 70.2 | 80.7 | 81.6 | 94.8 | 91.6 | 95.8 | 98.5 | 66.2 | 84.9 |
| Representation Surgery | 68.3 | 72.3 | 88.7 | 97.7 | 91.0 | 89.5 | 98.9 | 72.9 | 84.9 |
| *Date-free Methods* | | | | | | | | | |
| Weight Averaging | 67.7 | 70.0 | 75.3 | 79.5 | 74.9 | 60.1 | 94.4 | 43.8 | 70.7 |
| Fisher Merging | 68.5 | 69.9 | 75.2 | 80.4 | 73.2 | 61.2 | 94.5 | 50.7 | 71.7 |
| RegMean | 69.1 | 71.6 | 77.6 | 88.8 | 83.7 | 70.2 | 96.9 | 54.6 | 76.6 |
| Task Arithmetic | 61.1 | 65.9 | 74.0 | 76.2 | 88.0 | 73.9 | 98.4 | 53.0 | 73.8 |
| Ties-Merging | 69.1 | 72.5 | 80.5 | 84.0 | 85.0 | 71.5 | 98.1 | 54.9 | 77.0 |
| Consensus Merging | 69.8 | 71.4 | 80.8 | 86.5 | 88.0 | 71.1 | 98.4 | 57.0 | 77.9 |
| **WUDI-Merging** (ours) | **75.7** | **82.5** | **90.7** | **98.0** | **95.4** | **96.6** | **99.4** | **74.7** | **89.1**$_{\triangle 11.2}$ |

Table 11: Multi-task performance when merging RoBERTa models on 8-task GLUE benchmark. We report normalized score (Lu et al., 2024).

| Method | CoLA | SST-2 | MRPC | STS-B | QQP | QNLI | MNLI | RTE | Avg. |
|---|---|---|---|---|---|---|---|---|---|
| Pre-trained | 0.0 | 53.8 | 85.0 | 4.0 | 37.5 | 53.1 | 37.1 | 71.2 | 41.7 |
| Individual | 100.0 | 100.0 | 100.0 | 100.0 | 100.0 | 100.0 | 100.0 | 100.0 | 100.0 |
| Weight Averaging | 0.0 | 59.2 | 85.8 | 47.0 | 45.4 | 63.9 | 48.0 | 71.2 | 52.6 |
| Task Arithmetic | 8.4 | 88.3 | **89.6** | 32.8 | 82.0 | 85.4 | 75.5 | 80.4 | 67.8 |
| Ties-Merging | 31.8 | 88.9 | 86.2 | 10.9 | 61.1 | 85.9 | 83.0 | 69.6 | 64.7 |
| Task Arithmetic (w/ DARE) | 0.0 | 88.1 | 86.6 | 30.2 | 84.3 | 79.1 | 64.0 | 77.2 | 63.7 |
| Ties-Merging (w/ DARE) | 11.8 | 95.5 | 85.8 | 9.4 | 86.8 | 88.7 | 83.1 | 63.6 | 65.6 |
| **WUDI-Merging** (ours) | **81.8** | **98.3** | 78.7 | **60.5** | **92.7** | **90.5** | **93.3** | **86.4** | **85.3**$_{\triangle 19.7}$ |

Table 12: Multi-task performance when merging RoBERTa-Large models on 8-task GLUE benchmark. We report normalized score (Lu et al., 2024).

| Method | CoLA | SST-2 | MRPC | STS-B | QQP | QNLI | MNLI | RTE | Avg. |
|---|---|---|---|---|---|---|---|---|---|
| Pre-trained | 0.0 | 51.5 | 40.9 | 20.9 | 36.4 | 56.0 | 37.6 | 62.4 | 38.2 |
| Individual | 100.0 | 100.0 | 100.0 | 100.0 | 100.0 | 100.0 | 100.0 | 100.0 | 100.0 |
| Weight Averaging | 7.4 | 55.1 | 84.2 | 46.3 | 56.7 | 73.8 | 35.8 | 66.7 | 53.3 |
| Task Arithmetic | 7.4 | 86.1 | 86.8 | 78.0 | 90.7 | 77.0 | 73.3 | 67.6 | 70.9 |
| Ties-Merging | 42.7 | 78.1 | 85.2 | 51.7 | 89.9 | 81.9 | 79.7 | 70.0 | 72.4 |
| Task Arithmetic (w/ DARE) | 4.1 | 85.2 | 85.8 | 71.6 | 91.3 | 85.6 | 75.2 | 68.1 | 70.9 |
| Ties-Merging (w/ DARE) | 2.9 | 90.4 | 86.8 | 75.4 | 92.4 | 86.4 | 79.0 | 69.1 | 72.8 |
| **WUDI-Merging** (ours) | **82.2** | **98.7** | **87.3** | **81.4** | **94.6** | **96.6** | **93.4** | **77.1** | **88.8**$_{\triangle 16.0}$ |

### A.5. Consistency of the input between pretrained model and fine-tuned model

In this section, we provide the input consistency between the pretrained model and the expert models, as shown in Fig.8 and Fig.9. It can be seen that the large model shows less consistency, however the change of the direction and magnitude still very small, where the change of the direction of the most layer is less than 0.4. Consider that he task vector is the weighted sum of the input from the start to end during fine-tuning, the task vector has better a consistency than the consistency between the pretrain model and the fine-tuned model.

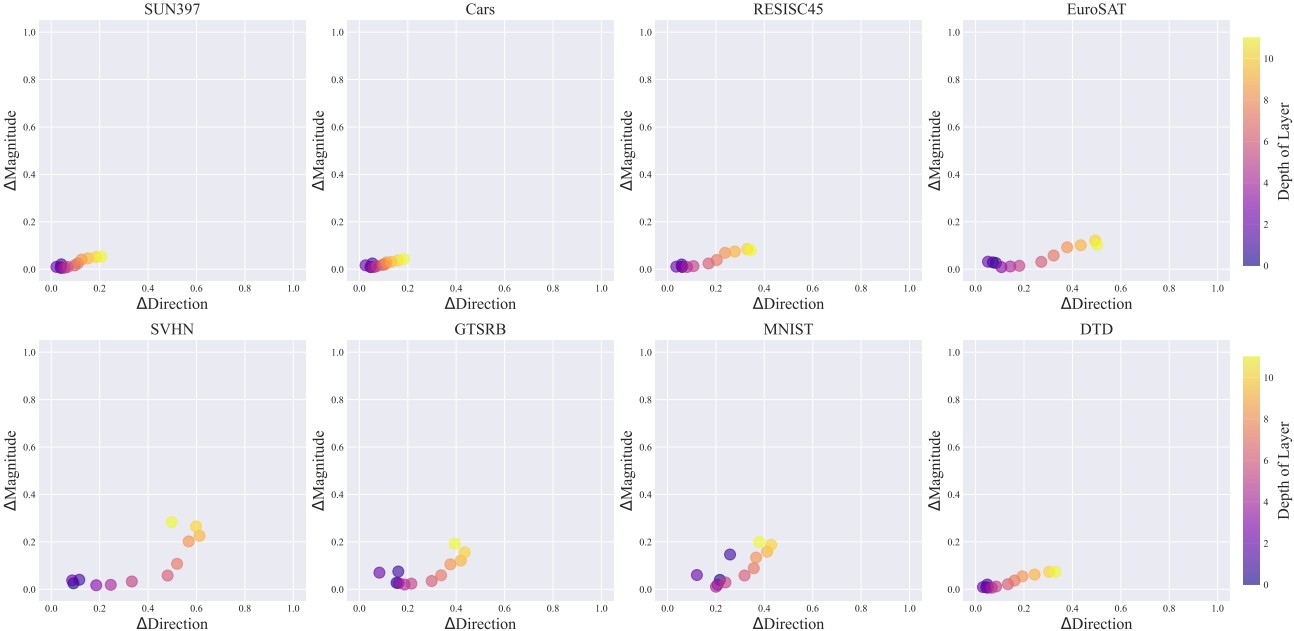

Figure 8: The input consistency between the pretrained model and the fine-tuned model for ViT-B/16

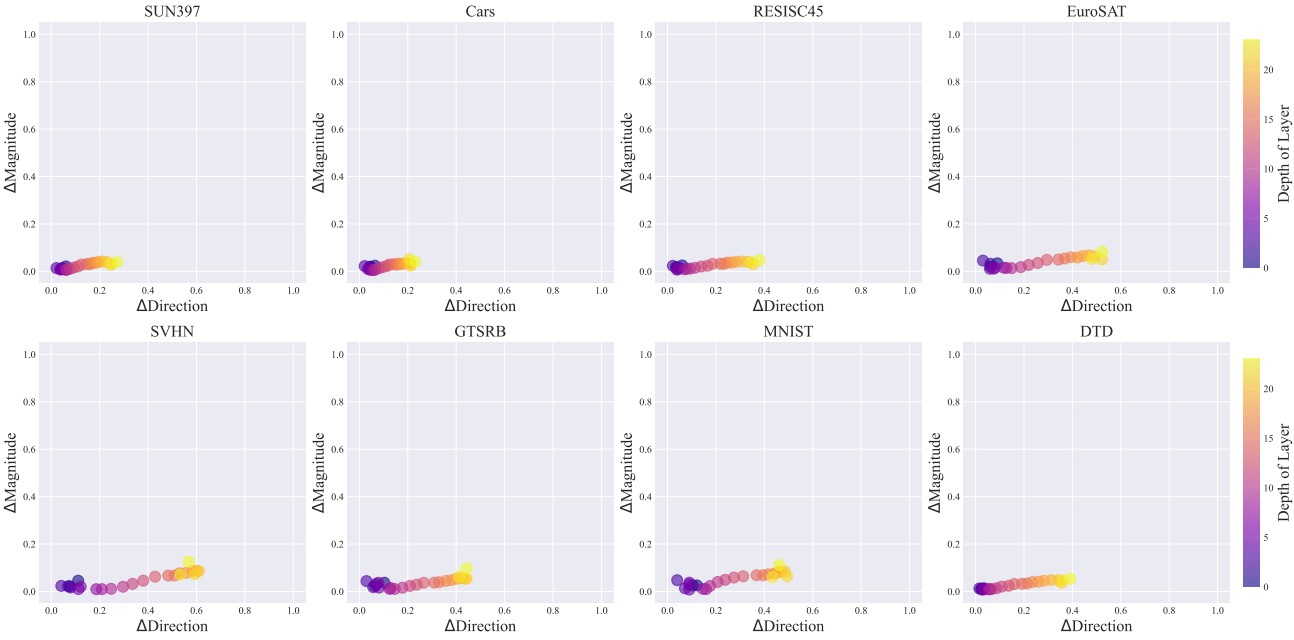

Figure 9: The input consistency between the pretrained model and the fine-tuned model for ViT-L/14

### A.6. The detailed results of interference in different layers and tasks.

To further demonstrate our method's capability to eliminate layer-wise interference, we use relative error for evaluation. Specifically, for a given merged vector $\boldsymbol{\tau}_m$, the relative error is calculated as follows:

$$\text{Relative Error} = \frac{1}{N} \sum_{n=1}^{N} \frac{||f(x_n; \theta + \boldsymbol{\tau}_m) - f(x_n; \theta + \boldsymbol{\tau}_i)||}{||f(x_n; \theta + \boldsymbol{\tau}_i)||} \tag{83}$$

Fig. 10, 11, and 12 respectively present the relative errors of different models with varying numbers of layers across different datasets. Fig. 3 summarizes the average relative errors across these datasets for the models with different layer counts.

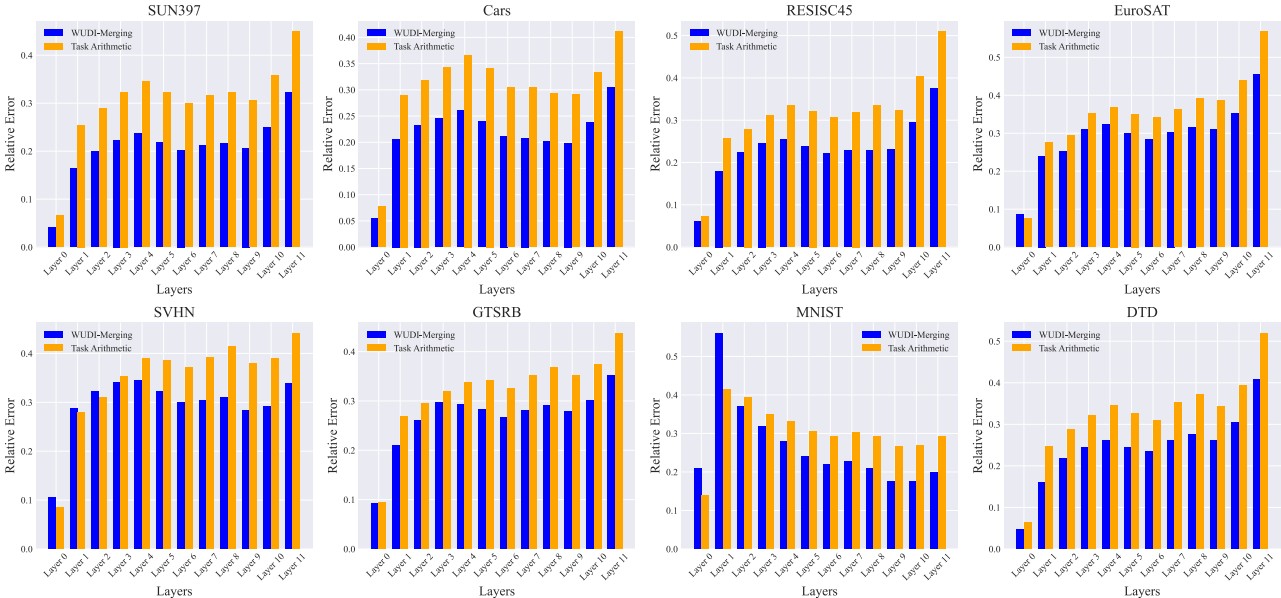

Figure 10: The interference for different layers and tasks for ViT-B/32.

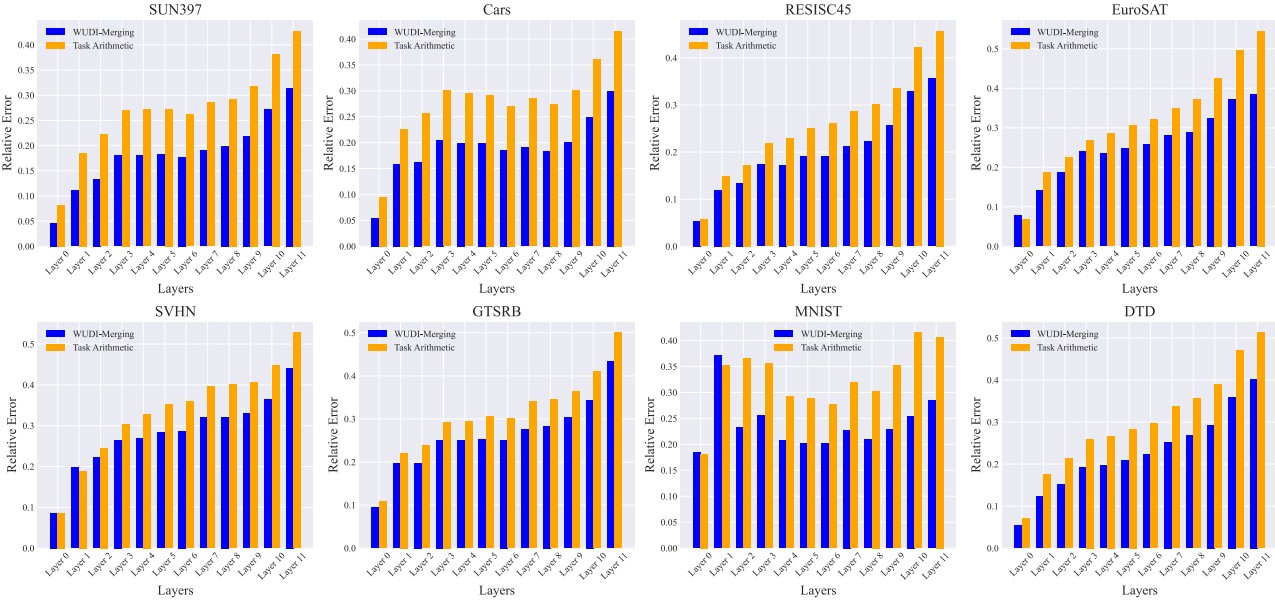

Figure 11: The interference for different layers and tasks for ViT-B/16.

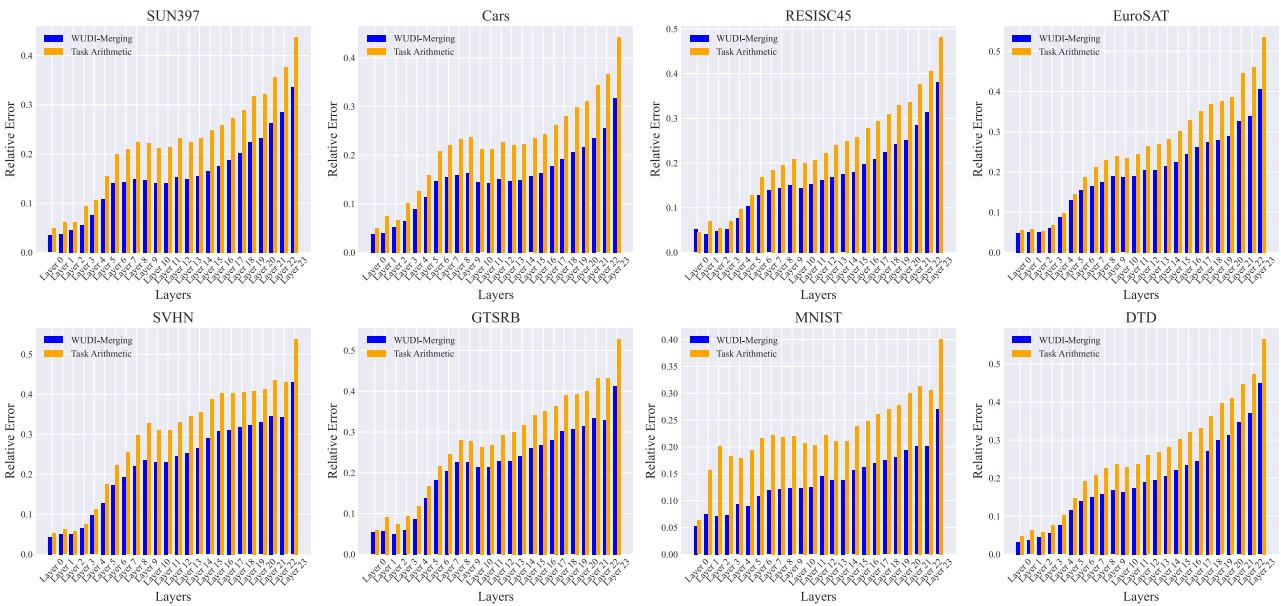

Figure 12: The interference for different layers and tasks for ViT-L/14.

### A.7. Performance comparison when using different steps for the solution process.

Fig. 13 presents performance comparison for models of various sizes, including ViT-B/32, ViT-B/16, and ViT-L/14, when using different steps for the optimization process.

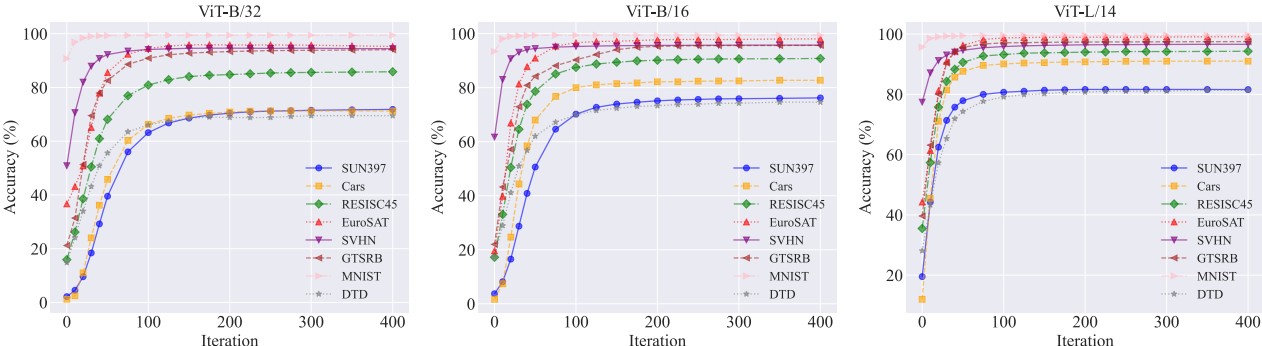

Figure 13: The results obtained at different stages of the solution process.

### A.8. Details of the experiment.

#### A.8.1. DATASETS

**Vision tasks**. Following (Ilharco et al., 2023; Yadav et al., 2023; Yang et al., 2024b), we use SUN397 (Cordts et al., 2016), Cars (Krause et al., 2013), RESISC45 (Cheng et al., 2017), EuroSAT (Helber et al., 2019), SVHN (Netzer et al., 2011), GTSRB (Stallkamp et al., 2012), MNIST (LeCun et al., 1998), DTD (Cimpoi et al., 2014)
**NLP tasks**. For *discriminative language tasks*, we use GLUE benchmark (Wang et al., 2018). For *generative language tasks*, we use AlpacaEval (Dubois et al., 2024), GSM8K (Cobbe et al., 2021), MATH (Hendrycks et al., 2021), HumanEval (Chen et al., 2021), MBPP (Austin et al., 2021).

#### A.8.2. CALCULATE RESOURCES AND ENVIRONMENT.

All of our experiments were conducted on NVIDIA A100 40GB, Python 3.10, PyTorch 2.4.0, and the CUDA 11.8 toolkit.

#### A.8.3. HYPERPARAMETERS

There are only two hyperparameters in our method. In practice, we set the learning rate to 1e-5 and the number of iteration to 300. Following (Jin et al., 2023; Xiong et al., 2024), we only applied our method to the linear layer in the model.

