# OpenReview forum: "Whoever Started the interference Should End It: Guiding Data-Free Model Merging via Task Vectors"
_ICML.cc/2025/Conference — ICML 2025 poster_

### Official Review · Reviewer_7sbW · 2025-02-20

**Overall Recommendation:** 2

**Summary:**

The paper offers very satisfying results and method in a nice writeup that could be communicated much better.

**Claims And Evidence:**

You refer to Task Arithmetic as data-free which is mostly true, except that it reweights the models with data. Classic merging does not and Ties also has a scaling factor but it is less sensitive to it so the default is more likely to be just fine.

**Essential References Not Discussed:**

The linearity claims remind me of works like https://arxiv.org/abs/2303.09435 https://arxiv.org/abs/2303.08112 from interpretability and of works like https://arxiv.org/abs/2407.15845 on reconstructing data from weights or of memebership attacks.

The claim in 140 should be referred to https://arxiv.org/abs/2302.04863, this is the paper that deals with this (while Task arithmetic is doing many nice things, this is not the main source of knowledge regarding that claim).

Maybe useful to cite the ones proposing model merging when you first use the term? I think you did cite (most?) of them somewhere else (https://arxiv.org/abs/2203.05482 https://arxiv.org/abs/2111.09832 https://arxiv.org/abs/2204.03044 )

The moerging lit. is not essential, but it is quite lacking (maybe arrow and phatgoose are the clearest cases to cite? there is also a survey if it helps (https://arxiv.org/abs/2408.07057)

**Experimental Designs Or Analyses:**

The baselines on NLP are outdated, RoBERTa is a better hyperparameter tuned (slightly larger) BERT, the first model to pretrain (parallel to Elmo), a lot has changed since.

Beyond this, the method performs extremely well, it is compared with a lot of baselines and (while for some reason not highlighted) almost reaches multitask learning, all without data, this is a massive result.

**Methods And Evaluation Criteria:**

The paper mainly focuses on multitask learning (not generalization, new capabilities etc.) somewhat awkwardly it focuses on task arithmetic quite heavily although this is not even the idea behind task arithmetic (there the idea is compositionality of tasks not orthogonality and lack of interference which are closer to the first merging works, to ties, to https://proceedings.neurips.cc/paper_files/paper/2023/file/d28077e5ff52034cd35b4aa15320caea-Paper-Conference.pdf etc.).

**Other Comments Or Suggestions:**

constitude-> constitute
"Crucially,
since the learning rate and total optimization steps always
remain constrained during standard fine-tuning procedures,
the corresponding inputs for individual samples remain consistent across successive update iterations." Can't parse this sentence, the inputs are consistent? In what sense?
l61 indicate"s"?
" As shown in Figure A.3" A.3 is an App.~ not a figure. Also the appendix has only subsections and no sections? why?
l137 (right col) what is p? probability over the distribution of inputs of the training data in i? all tasks? something else?
l160 (right) "lemma1" missing space same sentence "a individual" wrong det.
l179 "highly consistency"
l194 eq.1 missing space
l198 why does preposition has a . instead of a space? (there are many typos, you should pass over it, grammarly might find some of them as well) Also in Theorem.1, if the thing is that you want space and don't want it to jump lines, use ~ instead of a regular space (it is a space that counts as a regular character and not a word boundary)
l243 "the" unnecessary?
l256 (right) \citet not \cite or \citep when using the citation as part of the sentence.
l262 right I am not sure I would call it analysis, maybe just evaluation or performance? We don't learn anything, just compare.
What are the red delta numbers in the tables? Difference from what?

**Other Strengths And Weaknesses:**

The paper is generally clear, but there are many small missing pieces that are left for the reader to reinvent, this makes following the paper very hard. I tried to highlight those cases (what is an input, what distribution are we discussing etc.) and explain where is the confusion, but at the current state it looks really well when skimming but makes a really hard life for following along.

It is unclear from the figure\intro what do you mean as input vector. for an LLM there is the overall input for all of the training (but it is not a vector), the input to a given layer or the input to the whole network (looks like the second).

I do not understand the claim in l140 "This directional consistency suggests that task vectors converge to local optima". Local and not global? (that is obvious) optima and not just anywhere? Well this what makes convergence. Are you trying to say the local optima is the same and unique? This is only partially true (see Guetta et al. 2023 I also shared in the related work section). Moreover, the fact that models converge to similar solutions doesn't mean that the differences between them are noise (interference they are for sure not, as the literature thinks of interference as places where the different Task Vectors ruin for each other, regardless if it is good for the original task). By the definition of the delta, it seems like you are thinking of the interference as *any* change from the task vector, this is obviously not true, the whole point of task arithmetic was that you can "add" things together, merging methods also show cases where you can get disentangled vectors that do not interfere (although they do change) to some extent.

**Questions For Authors:**

In your method you relearn all of the weights, so in some essence you have no reason to fall in any convex hull of the original models (like AdaMerge for example). Is that an important feature of the method? Do you think that is the main gain?
More understanding of what this optimization procedure actually finds might be illuminating (both on what should be still improved, and what is being done here successfully).

Is the choice of RoBERTa related to what you could train yourselves? because if so, maybe this can help? https://huggingface.co/Lots-of-LoRAs this is quite a large set of LORA models all trained similarly, if you use it because the method doesn't work with LORAs, this is something you should state much more clearly.

What methods do you see as addressing the least similar problem to yours?
Your method starts from a merged model, what if you started from a better merged model (of a dissimilar method?) would that result in another boost in performance?

**Relation To Broader Scientific Literature:**

The paper states well its relations to previous merging methods. It explains well what are the main changed and difficulties in preexisting methods and covers many baselines.

I do wonder how does this method work on LoRA and or whether KnoTS would be irrelevant here or will prealigning be still useful even thought you learn all the weights, but it is just something I am curious about and not a strong lack the paper has.

You mention not using external data, and that others do not, but other methods do (especially task arithmetic). Usually, they use a validation set to tweak some sizes. (noteworthy and very recent is arxiv.org/pdf/2310.02575 in that regard)
It is unclear how exactly did you run those methods and what you did yourself (did you use some validation set? where?).
Also recent, but not as recent as the above is KnoTS, which I wonder if, because you are assuming all kinds of linear stuff is further improving your method like it does to ties. i.e. is it complementary or conflicting.

**Theoretical Claims:**

There are proofs (and I commented also on some things that were unclear to me). In general, this takes a meaningful part of the paper, but I don't feel this is a main contribution of the paper.

---

> ### Author Rebuttal · Authors · 2025-03-28
>
> We appreciate the reviewers’ valuable feedback and have addressed each point as follows:
> ### Concern 1: The Use of Validation Data
> 1. **Whether task arithmetic is data-free**: We summarize TA as a data-free method as it can select empirical parameters for merging without requiring data. Although it can be improved by seaching rescaling coefficient, it is reasonable to summarize TA as a data-free method.
> 2. **How we run other methods**: We run other methods in the same way as described in their paper and report the results accordingly in the paper if the experimental settings are same.
> 3. **Whether we use validation data: Our proposed method is entirely data-free**, and did not require any validation data. This is because our approach is rescale-free and solely relies on our data-free loss.
> ### Concern 2: Focus on Multi-Task Learning
> Our work focuses on model merging rather than model editing. In current model merging field, multi-task learning is the primary goal to pursue [1,2,3,4,6,7], where TA and task vector is a foundamental concept for analysis the interference and conflict[1,2,6] .
> ### Concern 3: Related References
> 1. **Linearity claims**: Our work focued on the linear layer but not linearizing the whole model. Therefore, these linearization works are not highly relevant to us.
> 2.  **Reconstructing data**：Our work used the reconstruction as an intermediate step in our derivation and did not use reconstruction data in our method. Therefore, the data reconstruction attacks works methoded are not highly relevant to us.
> 3.  **Other works**: We appreciate the references you provided and will incorporate their discussion in our revision. The conclusions and methods in these works do not significantly overlap with ours. Therefore, we consider them to be "Related References Not Discussed" rather than "Essential References Not Discussed"
> ### W1: The Claim of Local Optima and Definition of Interference Vector
> 1. **The Claim of local optima** What we want to express is that the task vectors for same task are at similar positions, which indicated that the models are fine-tuned to convergence, so we think they are optimized to local optimal. But these task vectors may not be the optimal solution in the entire parameter space, so using local optimal is a more rigorous statement. Also we have not stated that this local optimal is unique, and this has no significant connection with our method.
> 2. **The Definition of Interference Vector: We do not claim that any changes to task vector constitute interference**, we define the interference as the difference between outputs of merged model and expert model, which is composed of $\delta x$, if $\delta$ is orthogonal to $x$, then $\delta$ is not cause interference. Similar concepts are adopted in methods like Eq.7 in Alpha-Edit [5] and Eq.1 in Regmean[4]. Furthermore, the success of TA and other methods doesn‘t mean "you can get disentangled vectors that do not interfere" since the current model merging method is far from allowing model merging without reducing any performance.
> ### Q1: Convex Hull
> Not falling in convex hull is not unique to our method, as other methods do not constrain the sum of rescaling coefficients equal to 1. The reason why our method effective compared to previous data-free methods is our method implicitly leverages input data information via task vectors.
> ### Q2: The Choice of RoBERTa and LoRA Results
> We selected RoBERTa because it's a commonly used benchmark, as used in works such as [3]. We further provide results for other widely adopted LoRA benchmarks [1][2][3]. Due to character limitation, you can view the results in **Tab1.1 and Tab1.2 in our rebuttal to reviewer VcLV**.
> ### Q3: The Least Similar Problem to Our Work
> 1. **Least Similar Problem** Most existing methods focus on reducing the interference and conflicts [1,2,3,4,6,7], so it's hard to say what is the least similar problem to our work.
> 2. **Combine with other methods** Since our objective is a simple optimization problem, we think different initialization will only provide limited influence. The result in Tab.3.1 demonstrated that using different methods as initialization yields similar results.
>
> **Tab 3.1** Results of using different methods as initialization
> |Method|Ours|Ties+Ours|Adamerging+Ours|
> |-|-|-|-|
> |Acc.|85.2|84.9|85.0|
>
> ---
> We thank the reviewer again for their valuable feedback and hope our detailed responses address the concerns. We look forward to further discussion.
>
> [1] TIES-Merging: Resolving Interference When Merging Models
>
> [2] Parameter Competition Balancing for Model Merging
>
> [3] Twin-Merging: Dynamic Integration of Modular Expertise in Model Merging
>
> [4] Dataless Knowledge Fusion by Merging Weights of Language Models
>
> [5] AlphaEdit: Null-Space Constrained Knowledge Editing for Language Models
>
> [6] Task Singular Vectors: Reducing Task Interference in Model Merging
>
> [7] DELLA-Merging: Reducing Interference in Model Merging through Magnitude-Based Sampling

---

### Official Review · Reviewer_kYu3 · 2025-03-07

**Overall Recommendation:** 4

**Summary:**

This paper introduces WUDI-merging, a new data-free model merging method. The authors provide theory-backed idea that task vectors for a linear layer represent a linear subspace corresponding to its inputs. They use this knowledge to construct a merging method that aims to minimize the inference of the merged model's weights on their corresponding tasks. They also introduce a per-task weighting corresponding to the Frobenius norm of the task vector. When compared to other methods on a variety of benchmarks, WUDI-merge obtains a signficantly higher normalized average multi-task score than other data-free merging baselines and even test-time adaptation methods.

**Claims And Evidence:**

The claims are supported by evidence. The core observation, namely that task vectors for a linear layer represent a linear subspace corresponding to its inputs, and the derivation of the optimization problem are backed by theory. WUDI-merging is compared to a variety of data-free merging methods and TTA methods on a variety of benchmarks, where it achieves SoTA performance. Sufficient ablation studies are conducted on their method.

One potential caveat, however, is that experiments are missing that explicitly test some of the theoretical results in the paper. This would include seeing how close the inputs to a linear layer are to the subspace corresponding to its task vector (i.e. experimental exploration of the reconstruction error in equation (13)). The theoretical analysis, while sound, involves the use constants bounding the norm of gradients, so the bound on the error might be large if those constants are large. Since the main purpose of these theoretical results was to motivate the construction of WUDI-merging, this does not significantly detract from the main contribution of this paper.

**Essential References Not Discussed:**

None that I am aware of.

**Experimental Designs Or Analyses:**

I didn't review anything in detail, but the experimental designs and analysis makes sense to me.

**Methods And Evaluation Criteria:**

Evaluation of average normalized accuracy on multi-way merges for vision models, discriminative text models, and generative text models makes sense for evaluation of a merging method.

**Other Comments Or Suggestions:**

Typos:
- line 175, the "Where" should not be capitalized
- Table 2, "date-free" should be "data-free"
- line 994: "that he task vector" should be "that the task vector"

**Other Strengths And Weaknesses:**

Strengths:
- Theoretical results are used to motivate the method.
- Method is non-trivial and novel.
- Method is data-free and relatively computational cheap.
- Writing is clear.

Weaknesses:
- Computation times and memory requirements are not provided for baseline merging methods.

**Questions For Authors:**

In the "Selection of Linear Subspace" paragraph of section 4.4:
1. How exactly are a combination of task vectors and random vectors used to optimize the loss? Do you replace some neurons in the task vector with random vectors and run WUDI-merging as normal?
2. How are the random vectors selected?

It would be good to include this information in at least the appendix.

**Relation To Broader Scientific Literature:**

The paper categories existing merging methods into data-free, test-time adaptation, and MOE-like merging. The differences and advantages/disadvantages of these methods are explained, and the categorization of WUDI-merging as a data-free method makes sense.

**Theoretical Claims:**

I skimmed the proofs and did not find any issues. However, I did not examine them in detail.

---

> ### Author Rebuttal · Authors · 2025-03-30
>
> We appreciate the reviewers’ valuable feedback and have addressed each point as follows:
> ### W1: Reconstruction Error
>
> For calcuating the reconstruction error in Equation (13), we first obtain the input for each layer from a set of samples and then compute the reconstruction coefficients using the least squares method with its corresponding task vector. The reconstructed vector, $x_{\text{recon}}$, is derived using these coefficients along with the task vector. Then we calculate the Relative Reconstruction Error (RRE) for the sample from task $i$ as follows:
>
> $
> \text{Relative Reconstruction Error (RRE)} = \frac{\|x - x_{\text{recon}}\|}{\|x\|}, \quad \text{where}  \quad x_{\text{recon}} =\tau_i\(\tau_i^T\tau_i)^{-1}\\tau_i^T x .
> $
>
> Table 2.1 shows the result of different layers and tasks. which demonstrate that the relative reconstruction errors across different layers are extremely small, which further validates our theoretical analysis.
>
> **Table 2.1**: Reconstruction Error Results on eight tasks of different layers
>
> | Task    | SUN397   | Cars     | RESISC45 | EuroSAT | SVHN    | GTSRB   | MNIST   | DTD     |
> |---------|----------|----------|----------|---------|---------|---------|---------|---------|
> | Layer 1 | 1.3e-5   | 1.3e-5   | 1.3e-5   | 3.3e-3  | 7.1e-3  | 3.6e-3  | 5.8e-3  | 1.3e-5  |
> | Layer 3 | 1.1e-5   | 1.1e-5   | 1.2e-5   | 1.4e-5  | 1.6e-5  | 1.3e-5  | 1.3e-5  | 1.1e-5  |
> | Layer 6 | 1.0e-5   | 1.1e-5   | 1.1e-5   | 1.3e-5  | 1.2e-5  | 1.2e-5  | 1.2e-5  | 1.1e-5  |
> | Layer 12| 9.8e-6   | 1.1e-5   | 1.2e-5   | 1.1e-5  | 1.2e-5  | 1.1e-5  | 1.2e-5  | 1.0e-5  |
>
> ---
>
> ### W2: Resource Consumption Comparison
>
> We report the computational time and GPU memory usage of different method on ViT-B-32 tasks. In comparison to the Adamerging method, our approach not only improves performance but also significantly reduces computational cost. The details are summarized in the following table:
>
> **Table 2.2**: Detailed computational time and gpu memory requirements on ViT-B-32 tasks.
> | Method                        | Accuracy (%) | Time     | GPU Memory (GB) |
> |-------------------------------|--------------|----------|-----------------|
> | Ties Merging                  | 72.4         | 4 s      | 0               |
> | Adamerging                    | 81.1         | 127 min  | 17.1            |
> | WUDI-Merging-CFS (CPU)        | 84.4         | 5 s      | 0               |
> | WUDI-Merging-CFS (GPU)        | 84.4         | 2 s      | 1.8             |
> | **WUDI-Merging**              | **85.2**     | 1 min 54 s | 4.0           |
>
>
> ### Q1 & Q2: How to Use Random Vectors or Subsets of the Task Vector for Optimization
>
> **For the subset of task vectors**, we randomly sample a subvector from the original task vector as follows:
>
> $\tau^{\text{sub}}_i = \tau_i [\text{rand\\_index}, :]$
>
> The corresponding loss is computed as:
>
> $
> \mathcal{L}_{\text{sub}} = \sum\_{i=1}^{n} \frac{1}{\|\tau_i\|\_F^2}\ \delta_i(\tau^{\text{sub}}\_i)^\top = \sum\_{i=1}^{n} \frac{1}{\|\tau_i\|_F^2}\ (\tau_m - \tau_i) (\tau^{\text{sub}}_i)^\top$
>
>
> **For random vectors**, we sample from a Gaussian distribution whose mean and standard deviation are computed from the original task vectors:
>
> $
> \tau^{\text{random}}_i \sim \mathcal{N}(\mu_i, \sigma_i^2), \quad \text{where} \quad \mu_i = \text{mean}(\tau_i) \quad \text{and} \quad \sigma_i = \text{std}(\tau_i)
> $
>
> The loss in this case is given by:
>
> $
> \mathcal{L}_{\text{random}} =  \sum\_{i=1}^{n} \frac{1}{\|\tau_i\|_F^2}\delta_i(\tau^{\text{random}}_i)^\top =  \sum\_{i=1}^{n} \frac{1}{\|\tau_i\|_F^2}(\tau_m - \tau_i)(\tau^{\text{random}}_i)^\top
> $
>
> **All reported results are averaged over 5 sampling runs**. We thank the reviewers for their suggestions and will include these additional details in the revision.
>
> ---
> We thank the reviewer again for their constructive feedback and hope that our detailed responses address the concerns. We look forward to further discussion.

---

### Official Review · Reviewer_VcLV · 2025-03-12

**Overall Recommendation:** 4

**Summary:**

This paper proposes WUDI-Merging, a data-free model merging method where the merged model weights are optimized via SGD using the Adam optimizer. The optimization objective leverages the insight that task vectors form an approximate linear subspace of the corresponding input space. Additionally, the authors provide a closed-form alternative (WUDI-Merging-CFS) for scenarios with limited computational resources. Empirical evaluations demonstrate that WUDI-Merging achieves state-of-the-art performance across vision and NLP tasks and is rescaling-free.

## update after rebuttal

I would like to thank the authors for the rebuttal. While I think the performance of WUDI is impressive and the new results on Flan-T5 and Ewen LoRAs are reassuring, I believe authors should make a huge effort for improving the presentation of the paper. As I have read authors' responses to also other reviewers, I found some of the responses were not very informative or persuasive. For example, as for the linear claims, I think using Lipschitz constant to explain it is too superficial. Also, Reviewer 7sbW's questions about how authors run other methods that require data are valid, but not answered. WUDI would be a great addition to the community if these problems are addressed and I strongly encourage the authors to handle the requests properly.

**Claims And Evidence:**

Yes

**Essential References Not Discussed:**

N/A

**Experimental Designs Or Analyses:**

Both vision and language model merging experiments follow the standard designs of previous model merging work. The analyses in Section 4.2, 4.4 are sound and valid.

**Methods And Evaluation Criteria:**

Yes

**Other Comments Or Suggestions:**

- The analysis stems from the assumption/literature observation that "the task vector in the linear layer encapsulates most of the capabilities of the expert models. As shown in Figure A.3, an expert model utilizing only the task vector of the linear layer achieves performance comparable to that of the full expert model. Therefore, we primarily focus on the linear layers of the model." I assume this may be model-dependent observation and thus I am particularly interested in the case when using only the task vector of the linear layers is not enough? Will WUDI be less effective in those cases?
- While WUDI-Merging is described as hyperparameter-efficient, it would be helpful to provide a more detailed discussion on how different learning rates or optimization settings affect the quality of the merged model.
- Please proofread the manuscript once again as there are several typos/mistake throughout. e.g. line 133 "linear linear", line 993 "magnitude still very small", etc.

**Other Strengths And Weaknesses:**

### Strengths

- The optimization objective is theoretically motivated and can be efficiently approached using Adam within the current framework (e.g., PyTorch). Corresponding computation cost is discussed.
- An alternative solution, WUDI-Merging-CFS, is also provided for scenarios with limited GPU resources, along with a sensitivity analysis on regularization coefficient.
- WUDI-Merging is rescaling-free, as demonstrated in Figure 4 (b).
- A comprehensive benchmark evaluation on both vision and language models support the superiority claims of proposed method.
- The experiments are well-designed and analysis provided further understanding into WUDI-Merging.

### Weaknesses

- The experiment for generative language models is limited to merging three Llama2 models, which is relatively insufficient. Expanding the evaluation to include merging *more models*, and *more model architectures* (e.g., encoder-decoder, decoder-only) would provide a more comprehensive understanding of the general applicability and potential limitations of WUDI.
- Experimental study of "Input consistency" and results of model merging performance are based on fully fine-tuned models. The paper does not explore scenarios where task vectors are derived from PEFT methods (e.g., LoRA). Further analysis on this would provide a clearer understanding of method's applicability.
- The potential limitations of proposed method are not well-discussed.

**Questions For Authors:**

N/A

**Relation To Broader Scientific Literature:**

This work may inspire researchers to further explore SGD-based optimization strategies for data-free model merging, potentially shifting the focus from traditional closed-form solutions or heuristic-based approaches.

**Theoretical Claims:**

I have reviewed the theoretical claims in the main paper. Lemma 1 regards input consistency, Proposition 1 approximates task vectors as a linear combination of inputs, and Theorem 1 provides an upper bound on interference. The proofs appear to be correct.

---

> ### Author Rebuttal · Authors · 2025-03-29
>
> We appreciate the reviewers’ valuable feedback and have addressed each point as follows:
>
> ----
>
> ### W1: Results on More Models and LoRA
>
> To further demonstrate the generalizability of our method on different models and LoRA, we supplemented the experiments on Flan-T5-base and Qwen-14B. For merging LoRA, We first restore BA back into the original matrix $(\tau_i=B_iA_i)$ , then apply WUDI-Merging directly to $\tau_i$ to obtain $\tau_m$, then merging it into $\theta_{base}$.
> The experimental results obtained from merging Flan-T5-base (LoRA fine-tuned) models and Qwen-14B (LoRA fine-tuned) are shown in the table below:
>
> **Tab 1.1:** Experimental results of merging Flan-T5-base (LoRA fine-tuned) models on all eight tasks.
>
> | Method| CoLA | MNLI | MRPC | QNLI | QQP  | RTE  | SST2 | STSB | Avg.  |
> |-----|----|---|----|----|---|----|------|------|--|
> | **Individual**            | 69.1 | 82.7 | 85.5 | 90.9 | 84.0 | 84.4 | 92.9 | 87.4 | 84.6  |
> | **Ties-Merging**          | 68.3 | 56.3 | 79.4 | 89.8 | 83.7 | 79.4 | 91.6 | 71.2 | 77.5  |
> | **AdaMerging++** | 69.1 | 60.3 | 78.4 | 90.0 | 83.6 | 79.1 | 91.6 | 74.1 | 78.3  |
> | **WUDI-Merging(Ours)**                  | 68.6 | 79.0 | 77.7 | 87.2 | 83.1 | 75.8 | 93.2 | 85.0 | **81.2(+2.9)**  |
>
> **Tab 1.2:** Experimental results of merging Qwen-14B (LoRA fine-tuned) models on all four tasks.
> | Method | MMLU    | TruthfulQA | BBQ     | CNN-DailyMail | Avg.   |
> |---|---|--|---|---|---|
> | **Individual**| 68.35|53.34|93.53|19.46|58.67|
> | **Task Arithmetic**           | 67.56   | 52.33      | 78.38   | 20.54         | 54.70  |
> | **Ties-Merging (w/ DARE)**    | 69.38   | 52.03      | 81.06   | 15.91         | 54.62  |
> | **WUDI-Merging(Ours)**|  69.17 | 55.71 | 80.56 | 17.33 | **55.69(+0.99)**  |
>
> While there is a slight degradation relative to individual models, our method demonstrates SOTA performance on merging LoRA fine-tuned models.
>
> ----
>
> ### W2: Potential Limitation
>
> We think that merging heterogeneous models may be the potential Limitation, as such models might not adhere to the assumptions underlying WUDI-Merging and current merging methods. Addressing these challenges is a key direction for our future research.
>
> ----
>
> ### Q1: On Using Only the Task Vector of the Linear Layers
>
> In the context of homogeneous model merging, we think that **"using only the task vector derived from the linear layers is usually enough"**. Removing the nonlinear task vector from the fine-tuned model can be viewed as applying a small offset to a limited number of parameters. For a $C$-Lipschitz continuous model, the change in the output is bounded by:
>
> $
> ||f_{\theta+\tau_i}(x) - f_{\theta+\tau_i^{linear}}(x)||\le C·||\theta+\tau_i - \theta+\tau_i^{linear}|| = C·||\tau_i^{non-linear}||
> $
>
> Considering that the parameters of the nonlinear layer accounting for only a small fraction (Qwen-14B$\approx$0.007%, LLama3.1-8B$\approx$0.003%) and has a small offset. The affect brought by only using the task vector of the linear layer is small. Therefore, we think that using only the task vector derived from the linear layers is usually enough. In all our other experiments, we found that utilizing only the task vector of the linear layer achieves performance comparable to that of the full expert model, which also confirms this point.
>
> However, if the parameter's offset are large,our method may have limitations in use, which is also a major limitation in the current field of model merging [1]. Current model merging methods need to limit the change of fine-tuned expert models to base model in a relatively small range. We will explore this limitation in our future work.
>
> ----
>
> ### Q2: Results on Different Optimization Strategies
>
> We evaluate the effects of applying different optimizers and learning rates in our method, which is shown in Tab 1.3 and Tab 1.4. Tab 1.3 shows that Adam (85.2%) and SGD (85.1%) achieve similar performance, which suggest that our method is not sensitive to kind of the optimizers. Tab 1.4 demonstrate that a smaller learning rate can better ensure the stability of optimization and enhance the performance of results.
>
> **Tab 1.3:** Experimental results of applying different optimizers in our method, the average accuracy of ViT-B-32 are reported.
> | | Adam|SGD|
> |-|-|-|
> |Acc.| 85.2| 85.1 |
>
> **Tab 1.4:** Experimental results of applying different learning rate in our method, the average accuracy of ViT-B-32 are reported.
> | 1e-5|1e-4|1e-3|1e-2|Ada-Merging|
> |-|-|-|-|-|
> | 85.2 |  84.3  |  83.9 | 83.5| 80.9 |
>
>
> ----
>
> ### Q3: Typo
>
> We appreciate the reviewers' attention to detail and will correct the identified typographical errors in the final revision.
>
> ---
>
> We thank the reviewer again for their constructive feedback and hope that our detailed responses address the concerns. We look forward to further discussion.
>
> [1] Language Models are Super Mario: Absorbing Abilities from Homologous Models as a Free Lunch

---

### Decision · Program_Chairs · 2025-05-01

**Decision:**

Accept (poster)

**Comment:**

This paper got mixed reviews feedback, i.e., 2 accept and 1 weak reject rating.

Before rebuttal reviewers suggested the strengths of the papers are: 1) The optimization objective is theoretically motivated and can be efficiently approached using Adam within the current framework; 2) Theoretical results are used to motivate the method; 3) Method is non-trivial and novel; 4) Method is data-free and relatively computational cheap.

weaknesses are: 1) experimental results are limited; 2) Computation times and memory requirements are not provided for baseline merging methods; 3) as mentioned by reviewers 7sbW, writing needs to be improved otherwise it is hard to follow.

After rebuttal, all the reviewers confirmed they had read the authors' response and would update review if needed. In. the AC-reviewer discussion phase, reviewer VcLV mentioned most of their concerns are fixed, but some parts like the linear claim was not addressed. Also reviewers 7sbW suggested their concerns are not addressed.

Given these AC decided to give weak accept rating. But suggested authors carefully addressed reviewers' comments in the final version.